# Wnt5a signaling induced phosphorylation increases APT1 activity and promotes melanoma metastatic behavior

Rochelle Shirin Sadeghi[1], Katarzyna Kulej[2], Rahul Singh Kathayat[3], Benjamin A Garcia[2], Bryan C Dickinson[3], Donita C Brady[1]*, Eric S Witze[1]*

[1]Department of Cancer Biology, Perelman School of Medicine, University of Pennsylvania, Philadelphia, United States; [2]Department of Biochemistry and Biophysics, Perelman School of Medicine, University of Pennsylvania, Philadelphia, United States; [3]Department of Chemistry, University of Chicago, Chicago, United States

**Abstract** Wnt5a has been implicated in melanoma progression and metastasis, although the exact downstream signaling events that contribute to melanoma metastasis are poorly understood. Wnt5a signaling results in acyl protein thioesterase 1 (APT1) mediated depalmitoylation of pro-metastatic cell adhesion molecules CD44 and MCAM, resulting in increased melanoma invasion. The mechanistic details that underlie Wnt5a-mediated regulation of APT1 activity and cellular function remain unknown. Here, we show Wnt5a signaling regulates APT1 activity through induction of APT1 phosphorylation and we further investigate the functional role of APT1 phosphorylation on its depalmitoylating activity. We found phosphorylation increased APT1 depalmitoylating activity and reduced APT1 dimerization. We further determined APT1 phosphorylation increases melanoma invasion in vitro, and also correlated with increased tumor grade and metastasis. Our results further establish APT1 as an important regulator of melanoma invasion and metastatic behavior. Inhibition of APT1 may represent a novel way to treat Wnt5a driven cancers.

DOI: https://doi.org/10.7554/eLife.34362.001

**\*For correspondence:**
bradyd@pennmedicine.upenn.edu (DCB);
ewitze@upenn.edu (ESW)

## Introduction

The Wnt5a signaling pathway plays a critical role in multiple biological processes as it regulates cell polarity and polarized cell movement during embryonic development. Wnt5a expression is strongly correlated with melanoma progression and metastasis (*Bittner et al., 2000*; *Carr et al., 2003*). Melanoma is the deadliest form of skin cancer, with a 5 year survival rate of only 17% for metastatic melanoma patients (*Sandru et al., 2014*). Previously, the elevated expression of Wnt5a in human melanoma samples and subsequent in vitro studies have suggested a role for Wnt5a in melanoma metastasis (*Bittner et al., 2000*; *Carr et al., 2003*; *Da Forno et al., 2008*). Importantly, Wnt5a expression is positively correlated with poor outcome, with high Wnt5a expression being associated with decreased patient survival (*Da Forno et al., 2008*). With this information, it has been suggested that Wnt5a expression be used as a prognostic clinical risk factor (*Da Forno et al., 2008*). However, the exact mechanisms by which Wnt5a promotes metastatic behavior remains unknown. Thus, elucidation of the signaling events downstream of Wnt5a that promote polarized cell movement and metastatic behavior have the potential to enhance our understanding of the contribution of Wnt5a to melanoma metastasis.

Our lab and others have shown Wnt5a signaling promotes migration and invasion in melanoma cells (*Dissanayake et al., 2007*; *Wang et al., 2015*; *Weeraratna et al., 2002*). Wnt5a is a secreted

ligand in the noncanonical Wnt signaling pathway and upon interacting with its Frizzled receptor, activates a signal transduction cascade (*Sato et al., 2010*). Wnt5a signaling controls cell polarity and directional cell movement, and Wnt5a treatment promotes recruitment and reorganization of proteins involved in cell adhesion and cell signaling (*Witze et al., 2008*; *Wang et al., 2015*). Wnt5a stimulation promotes asymmetrical localization of cell adhesion molecules, increases cell motility, and elevates levels of free $Ca^{2+}$ to induce $Ca^{2+}$ signaling in a polarized manner (*Witze et al., 2008*; *Witze et al., 2013*). The tyrosine kinase receptor ROR2, which is predominantly expressed in metastatic melanoma, has been shown to specifically interact with Wnt5a and mediate Wnt5a signaling (*O'Connell et al., 2010*). In addition to reorganization of the actin cytoskeleton and polarized localization of cell adhesion molecules, others have shown that elevated Wnt5a expression promotes melanoma invasion and human biopsies directly correlated Wnt5a expression with tumor grade (*Weeraratna et al., 2002*).

Pro-metastatic cell adhesion molecules, such as transmembrane receptors CD44 and MCAM (Melanoma Cell Adhesion Molecule), have been shown to play a role in cell migration and metastasis. CD44 levels are elevated in multiple cancer types and MCAM is expressed in melanoma cells and not untransformed melanocytes (*Wielenga et al., 1993*; *Shih et al., 1994a*; *Shih et al., 1994b*). Palmitoylation of these cell adhesion molecules strongly influences protein function. CD44 has been shown to be palmitoylated and inhibiting CD44 palmitoylation through cysteine point mutations strongly enhanced breast cancer migration (*Babina et al., 2014*). Similarly, cysteine point mutations that block MCAM palmitoylation increases metastatic cell behavior in melanoma cells both in vitro and in vivo (*Wang et al., 2015*).

S-Palmitoylation is the reversible addition of a 16-carbon fatty acid to cysteine residues via a thioester linkage. Palmitate addition is mediated by palmitoyltransferases and acyl protein thioesterases (APTs) remove palmitate from proteins to regulate protein localization, trafficking, and cell signaling (*Vartak et al., 2014*; *Linder and Deschenes, 2007*; *Lin and Conibear, 2015*). Acyl protein thioesterases 1 and 2 (APT1 and APT2; aka LYPLA1 and LYPLA2) are cytosolic depalmitoylases that have been proposed to act constitutively to depalmitoylate proteins from membranes. However, there is evidence that protein depalmitoylation is a signal regulated modification. Wnt5a stimulation has been shown to induce the depalmitoylation of both MCAM and CD44 through APT1 (*Wang et al., 2015*). Wnt5a mediated melanoma cell invasion is reduced by treatment with a small molecule inhibitor of APT proteins. Another group has also shown that growth factor stimulation is also sufficient to inhibit depalmitoylating activity of APT proteins (*Kathayat et al., 2017*). Wang, et al. demonstrated that increased APT1 expression increased melanoma invasion in the absence of exogenous Wnt5a (*Wang et al., 2015*). It remains to be determined how Wnt5a signaling increases depalmitoylation through APT1. These results led us to investigate the link between Wnt5a signaling and regulation of APT1-mediated depalmitoylation. We hypothesized APT1 depalmitoylating activity could be regulated by upstream signaling events, specifically the Wnt5a signaling pathway.

Here, we uncover a mechanism by which the non-canonical Wnt signaling pathway increases the depalmitoylating activity of APT1. We have mapped a regulatory phosphorylation site on APT1 and utilizing site specific point mutations, we unravel the impact of phosphorylation on APT1 activity and its importance in melanoma cell invasion. Using a depalmitoylation probe (DPP) that measures depalmitoylating activity (*Kathayat et al., 2017*; *Qiu et al., 2018;* and *Beck et al., 2017*), we show APT1 phosphorylation increases its depalmitoylating activity. This is the first study showing a signaling pathway can direct protein depalmitoylation through a phospho-switch on APT1 that we propose functions to disrupt an inactive APT1 dimer.

## Results

### Wnt5a signaling induces APT1 phosphorylation

Wnt5a induced metastatic cell behavior was previously shown to be dependent on increased protein depalmitoylation driven by APT1 (*Wang et al., 2015*). We hypothesized a mechanism for increased depalmitoylation could be a result of regulatory post-translational modifications of APT1 that increases APT1 activity. To investigate if APT1 is post-translationally modified in response to Wnt5a stimulation, we used mass spectrometry to identify Wnt5a dependent post-translational modifications (PTMs) of APT1. Low Wnt5a producing WM239A melanoma cells expressing CFP-FLAG tagged

wild type APT1 (APT1$^{WT}$-CFP-FLAG) were treated with purified Wnt5a and APT1 was immunoprecipitated using anti-FLAG magnetic particles. APT1 was eluted with FLAG peptide. The immunopurified APT1 was alkylated and trypsinized and analyzed by mass spectrometry. The only phosphorylation sites identified on APT1 were on serine residues 209 and 210. Whereas single phosphorylation of either serine 209 or 210 was detected, we can't rule out the possibility of dual phosphorylation (*Figure 1A and B*). These findings are consistent with a previous high throughput mass spectrometry screen by another group that identified APT1 phosphorylation on serine residues 209 and 210 in laryngeal cancer cells (Phosphosite.org, site group ID: 25299324 and 25299327).

We generated separate phospho-specific antibodies to phospho-site Ser209 and Ser210 of APT1 (YenZym Antibodies) to detect phosphorylated APT1 in melanoma cell lysates. To validate the antibodies, WM239A cells ectopically expressing APT1$^{WT}$-CFP-FLAG were treated with purified Wnt5a for 15 min. APT1$^{WT}$-CFP-FLAG was then immunoprecipitated and separated by SDS-PAGE followed by immunoblotting. With phospho-APT1 antibodies to both serine 209 and serine 210, we detected phosphorylated APT1 (pS209-APT1/pS210-APT1) that increased following Wnt5a stimulation (*Figure 1C*). The induction of APT1 phosphorylation at Ser210 was detected in whole cell lysate without immunoprecipitation, showing a relatively high stoichiometry of phosphorylated APT1 to total APT1 after Wnt5a stimulation (*Figure 1D*). We also observe a similar, but weaker, increase in exogenous phospho-APT1 in whole cell lysate utilizing the pS209-APT1 antibody in response to Wnt5a (data not shown). However, we were unable to detect phosphorylation of the endogenous APT1 in the whole cell lysate from these cells (data not shown). To determine the temporal kinetics of Wnt5a induced APT1 phosphorylation, WM239A cells ectopically expressing APT1$^{WT}$-CFP-FLAG were treated with purified Wnt5a for increasing lengths of time, APT1$^{WT}$-CFP-FLAG was then immunoprecipitated and APT1 phosphorylation was measured by immunoblotting. APT1 phosphorylation peaked at 30 min and began to decrease at 60 min (*Figure 1E*). To determine the function of the Wnt5a induced phosphorylation, a phospho-deficient APT1 double mutant was generated by mutating both serine residues 209 and 210 to alanine (from here on referred to as APT1$^{SA}$). When WM239A cells ectopically expressing APT1$^{SA}$ are treated with Wnt5a, the levels of phosphorylation of APT1 at Ser209 measured by immunoblotting were unchanged compared to cells expressing APT1$^{WT}$ (*Figure 1F*).

The crystal structure of APT1 has been previously determined (*Devedjiev et al., 2000*; *Won et al., 2016*) allowing us to model where the phosphorylated serine residues are positioned on the three-dimensional structure (PDB:5sym). We found that the phosphorylated serine residues 209 and 210 are solvent exposed on the lip of the hydrophobic putative acyl-binding channel and adjacent to the catalytic triad (Ser119, His208, Asp174), suggesting phosphorylation of these sites could influence APT1 activity (*Figure 1G*). Together, these results indicate Wnt5a signaling in melanoma cells induces APT1 phosphorylation at serine residues 209 and 210 and our phospho-specific antibodies are selective for these sites.

## APT1 phosphorylation increases depalmitoylating activity

To investigate how Wnt5a induced phosphorylation of APT1 affects its function, phosphorylated serine 209 was mutated to a negatively charged aspartic acid (APT1$^{S209D}$) to mimic the negative charge of phosphate on serine 209. To measure APT1 activity, we utilized a recently developed fluorescent depalmitoylation probe, DPP-3, that contains a thiol conjugated seven carbon fatty acid that when hydrolyzed, generates a fluorescent product that can be measured at $\lambda_{ex}$490/9 nm; $\lambda_{em}$545/20 nm (*Kathayat et al., 2017*). This probe served as a reporter to measure the depalmitoylating activity of APT1 and APT1 mutants both in vitro and in live cells. 6x His-tagged APT1 expressed and purified from *E. coli* was incubated with DPP-3 and the relative fluorescence was measured over time. APT1$^{S209D}$ was found to have increased depalmitoylating activity compared to APT1$^{WT}$ (*Figure 2A*). A catalytically inactive mutant where catalytic serine 119 is mutated to alanine (referred to as APT1$^{S119A}$) was used as a negative control and generated minimal fluorescence throughout the duration of the assay (*Figure 2A*). We next evaluated the depalmitoylating activity of APT1$^{WT}$ and APT1$^{S209D}$ using DPP-3 in the presence of increasing concentrations of DPP-3 substrate to determine the initial velocities at multiple substrate concentrations (*Figure 2B and C*). We measured higher initial velocities for APT1$^{S209D}$ at each substrate concentration compared to APT1$^{WT}$ (*Table 1*).

To determine the effect of Wnt5a stimulation on APT1 depalmitoylating activity, WM239A cells were incubated with DPP-3 and fluorescence emission was measured at 15, 30, and 45 min post

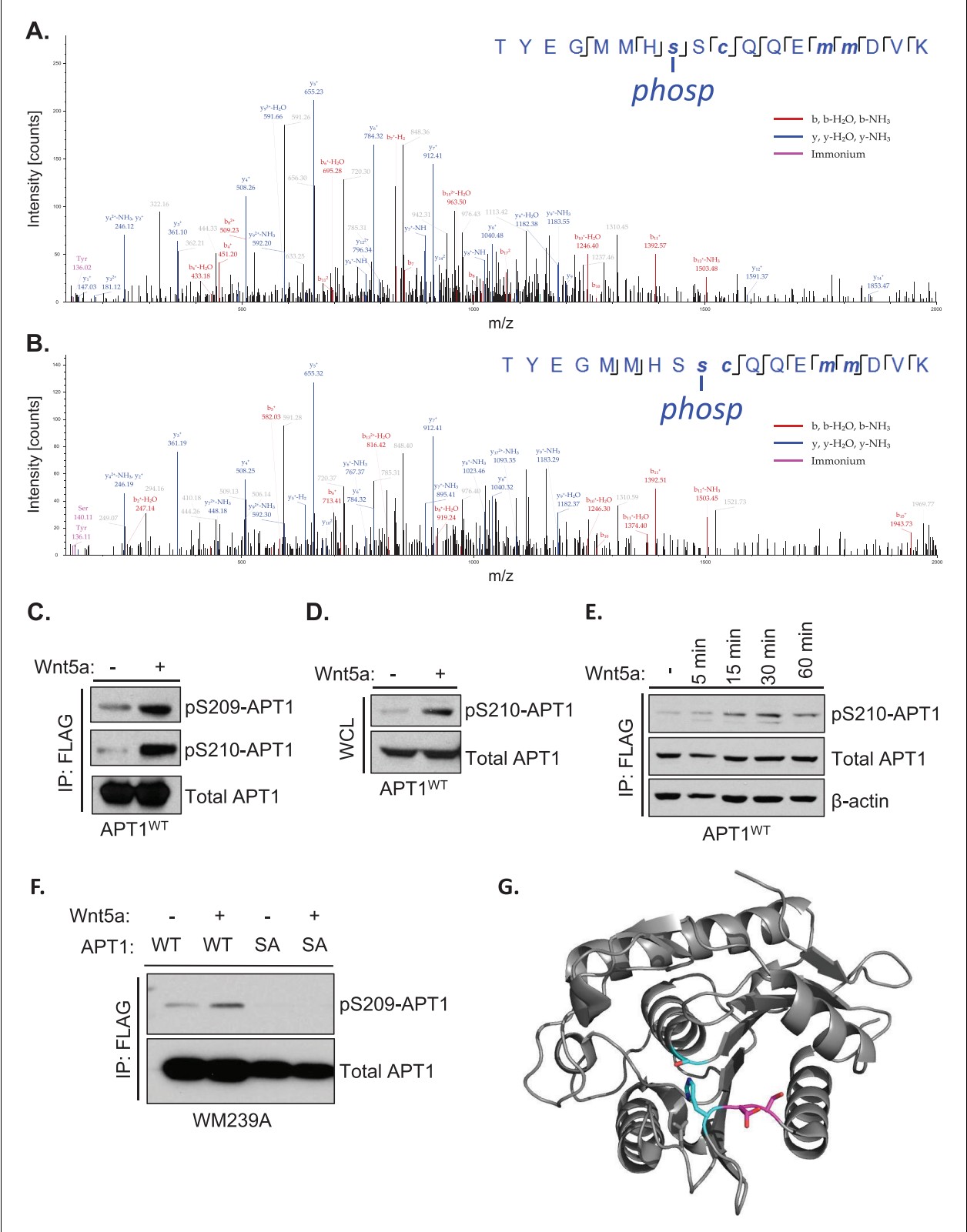

**Figure 1.** Wnt5a signaling induces APT1 phosphorylation on serines 209 and 210. (**A**) Annotated MS/MS spectrum of identified S209 phosphopeptide of protein APT1. The image represents the observed fragment ions collected using MS/MS (HCD). Colored lines represent matches between observed and expected fragment ions of the given peptides. Specifically, blue lines represent matches with y-type fragments, red lines with b-type fragments, and purple lines show immonium ions. (**B**) Annotated MS/MS spectrum of identified S210 phosphopeptide of protein APT1, as described in A. (**C**)

*Figure 1 continued on next page*

Figure 1 continued

APT1$^{WT}$-CFP-FLAG was immunoprecipitated from lysates prepared from WM239A melanoma cells expressing APT1$^{WT}$-CFP-FLAG treated with control buffer or 150 ng/ml of purified Wnt5a for 15 min and analyzed by SDS-PAGE followed by immunoblotting to detect phosphorylated APT1 using antibodies to pS209-APT1 and pS210-APT1. (D) Western blot of whole cell lysate (WCL) from WM239A cells expressing APT1$^{WT}$-CFP-FLAG treated with control buffer or 150 ng/ml of Wnt5a for 15 min. (E) WM239A cells expressing APT1$^{WT}$-CFP-FLAG were treated with control buffer or 150 ng/ml of Wnt5a for increasing lengths of time and then APT1$^{WT}$-CFP-FLAG was immunoprecipitated and analyzed by SDS-PAGE and immunoblotting to detect phosphorylated APT1 with anti-p210-APT1 antibody. (F) WM239A cells expressing APT1$^{WT}$-CFP-FLAG (WT) or APT1$^{SA}$ -CFP-FLAG (SA) were treated with control buffer or 150 ng/ml of Wnt5a for 15 min. APT1 was immunoprecipitated and analyzed by SDS-PAGE and immunoblotting to detect phosphorylated APT1 with anti-pS209-APT1 antibodies. (G) Three-dimensional model of APT1, showing the positions of Ser209 and Ser210 (magenta) adjacent to the catalytic residues Ser119, Asp174 and His208 (cyan).

DOI: https://doi.org/10.7554/eLife.34362.002

Wnt5a treatment by live-cell microscopy (*Figure 2—figure supplement 1A*). We quantified fluorescence emission in individual cells and determined that Wnt5a stimulation significantly increased the fluorescence emission compared to control buffer treated cells, indicating an increase in endogenous APT1 activity (*Figure 2D*). To confirm the increase in fluorescence emission in response to Wnt5a was through APT1, we immunoprecipitated APT1 from WM239A APT1$^{WT}$ cells stimulated with purified Wnt5a or untreated control cells, and measured depalmitoylating activity in vitro using DPP-3. APT1 isolated from Wnt5a stimulated cells showed an increase in depalmitoylating activity compared to untreated control (*Figure 2—figure supplement 1B*). While the non-canonical Wnt ligand Wnt5a is sufficient to increase APT1 activity we found stimulation with the canonical Wnt ligand, Wnt3a, was unable to increase APT1 activity compared to Wnt5a and was similar to cells treated with control buffer (*Figure 2E*). We confirmed these cells are competent to respond to the Wnt3a concentration used because we observe stabilization of β-catenin beginning at 75 ng/ml of Wnt3a (*Figure 2—figure supplement 1C*). Therefore, the ability to increase APT1 activity appears to be specific to the non-canonical Wnt pathway.

To directly assess how the phosphorylation state of APT1 affects depalmitoylation in live cells, WM239A melanoma cells expressing APT1$^{WT}$ or the phospho-mutants APT1$^{SA}$ and APT1$^{S209D}$ were incubated with the DPP-3 probe for 30 min and fluorescence emission was measured by live-cell microscopy. Consistent with the in vitro results, APT1$^{S209D}$ has greater depalmitoylating activity in cells compared to either APT1$^{WT}$ or APT1$^{SA}$ (*Figure 2F*, *Figure 2—figure supplement 1D*). To determine the contribution of APT1 phosphorylation to the Wnt5a mediated increase in APT1 activity, we treated cells expressing either APT1$^{WT}$, APT1$^{SA}$, or APT1$^{S209D}$ with Wnt5a or control buffer and measured the APT1 activity with the DPP-3 probe. Wnt5a treatment increases APT1$^{WT}$ activity to levels higher than the phosphomimetic mutant APT1$^{S209D}$. As expected Wnt5a stimulation is unable to increase activity of phospho-dead APT1$^{SA}$ or further increase the constitutive elevated activity of APT1$^{S209D}$ (*Figure 2F*).

We next asked if inhibiting protein phosphorylation pharmacologically was sufficient to block the Wnt5a mediated increase in APT1 activity. Because the identity of the kinase that phosphorylates APT1 is still unknown, we pre-treated WM239A cells expressing APT1$^{WT}$ with either the serine/threonine kinase ATP-competitive inhibitor BI-D1870 or the broad-spectrum kinase inhibitor staurosporine for 1 hr, incubated the cells with the DPP-3 probe, and measured fluorescence over time by live-cell microscopy during treatment with Wnt5a or control buffer. Treatment with either protein kinase inhibitors inhibited the Wnt5a mediated increase in APT1 activity to levels comparable to unstimulated cells (*Figure 2G*). These results suggest that phosphorylation of APT1 is a main driver of increased APT1 activity in response to Wnt5a.

## Expression of APT1$^{S209D}$ decreases levels of a palmitoylated protein substrate

We next asked if we could observe a similar phosphorylation dependent increase in depalmitoylating activity by measuring palmitoylation of a protein substrate in cells. The extent of protein palmitoylation is the result of an equilibrium between palmitoylation and depalmitoylation of substrates. To determine the immediate effect of expressing the phospho-mutant forms of APT1 on protein palmitoylation before the cell adapts and reaches a new steady state, we generated an inducible expression system to measure the effects of acute expression of APT1$^{WT}$ and the APT1 phosphorylation

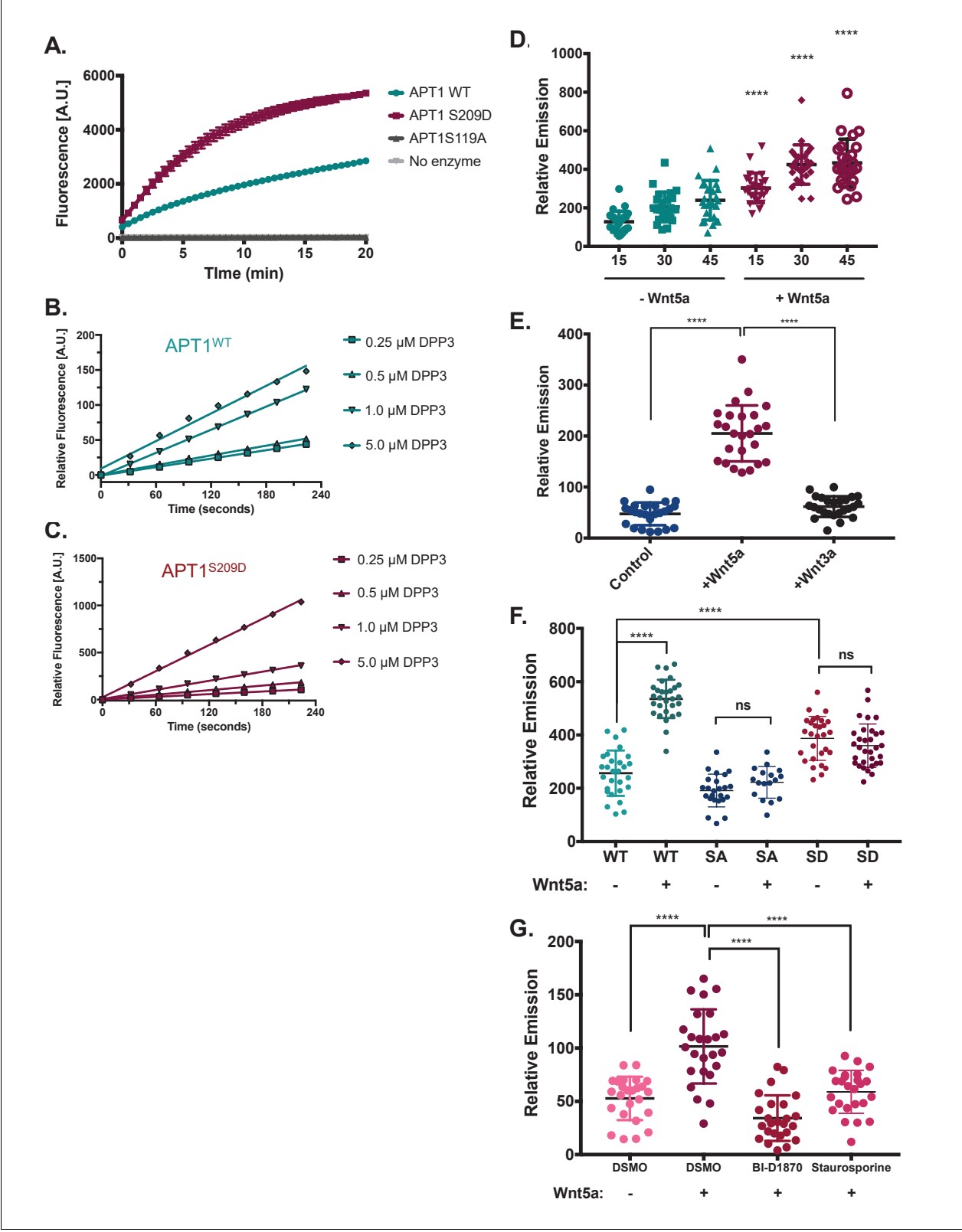

**Figure 2.** Wnt5a stimulation increases APT1 depalmitoylating activity through phosphorylation. (A) In vitro fluorescence assays using 5 µM of depalmitoylation probe DPP-3 in HEPES (20 mM, pH 7.4, 150 mM NaCl, 0.1% Triton X-100) with either 50 nM purified APT1$^{WT}$, APT1$^{S209D}$ or catalytically inactive APT1$^{S119A}$ and fluorescence emission was measured over time ($\lambda_{ex}$490/9 nm; $\lambda_{em}$545/20 nm). Error bars indicate s.e.m., n = 3 replicates per condition. Results shown are from a representative trial from three independent experiments. (B) Linear regression of APT1$^{WT}$ (WT) enzymatic activity

*Figure 2 continued on next page*

*Figure 2 continued*

measured by fluorescence emission of increasing DPP-3 substrate concentrations over time. Results are averaged from six independent experiments.
(C) Linear regression of APT1[S209D] (SD) enzymatic activity measured by fluorescence emission of increasing DPP-3 substrate concentrations over time.
Results are averaged from six independent runs. (D) Quantification of relative fluorescence of WM239A cells treated with control buffer or 150 ng/ml of
Wnt5a, loaded with 10 µM DPP-3, and then analyzed by live-cell fluorescence microscopy over time. Error bars indicate s.d., n = 25 cells per condition,
****$p < 0.0001$ by unpaired t-test analysis. Results shown are from three experiments. (E) Quantification of relative fluorescence of WM239A cells
expressing APT1[WT] treated with control buffer, 150 ng/ml of Wnt5a, or 150 ng/ml of Wnt3a, loaded with 5 µM DPP-3, and then analyzed by live-cell
fluorescence microscopy after 30 min. Error bars indicate s.d., n = 25 cells per condition, ****$p < 0.0001$ by unpaired t-test analysis. Results shown are
from three experiments. (F) Quantification of relative fluorescence generated by WM239A cells expressing either APT1[WT] (WT), APT1[SA] (SA), or
APT1[S209D] (SD) treated with control buffer or 150 ng/ml Wnt5a, loaded with 10 µM DPP-3, and then analyzed by live-cell fluorescence microscopy after
30 min. Error bars indicate s.d., n = 17–32 cells per condition, ****$p < 0.0001$ by unpaired t-test analysis. Results shown are from three experiments. (G)
Quantification of relative fluorescence of WM239A APT1[WT] cells treated with kinases inhibitors 10 µM BI-D1870, or 0.2 µM staurosporine or DMSO
control for 1 hr, loaded with 10 µM DPP-3, treated with Wnt5a and analyzed by live-cell fluorescence microscopy after 30 min. Error bars indicate s.d.,
n = 25 cells per condition, ****$p < 0.0001$ by unpaired t-test analysis. Results shown are from three experiments.Quantification for all live-cell microscopy
was determined by measuring the mean intensity of relative fluorescence for region of interests (background fluorescence was subtracted from mean
intensity).

DOI: https://doi.org/10.7554/eLife.34362.003

The following figure supplement is available for figure 2:

**Figure supplement 1.** Activity of APT1 in response to Wnt5a stimulation or the phosphomimetic mutation.
DOI: https://doi.org/10.7554/eLife.34362.004

mutants on a specific depalmitoylation substrate. We chose MCAM (melanoma cell adhesion molecule), a known substrate of APT1 that is depalmitoylated in response to Wnt5a stimulation, to investigate the contribution of APT1 phosphorylation on protein depalmitoylation (*Wang et al., 2015*). HEK 293T cells were co-transfected with a MCAM-GFP plasmid and a doxycycline inducible plasmid containing FLAG-tagged APT1[WT], APT1[SA], or APT1[S209D]. Cells expressing APT1[WT] or mutants were induced with doxycycline for 15 hr. We found cells produced similar amounts of wild type and mutant APT1 protein (*Figure 3A*). The abundance of palmitoylated MCAM in each condition was then measured using the acyl biotin exchange assay (ABE). As a negative control, samples are processed without hydroxylamine (-HAM) leaving the palmitoylated cysteines intact and preventing biotin labeling. We found MCAM palmitoylation to be highest in APT1[SA] samples, indicating a decreased ability to depalmitoylate MCAM (*Figure 3B*). In contrast, the APT1[S209D] sample had the lowest level of palmitoylated MCAM, indicating the highest level of depalmitoylating activity. These results confirm the elevated depalmitoylating activity of APT1[S209D] we observe with the DPP-3 probe using a known palmitoylated protein in the cell. These data serve as evidence that the phosphorylation state of APT1 regulates depalmitoylation of endogenous substrates.

**Table 1.** Enzyme kinetics of APT1[WT] and APT1[S209D].
Table of initial velocities ($V_0$) of APT1[WT] and APT1[S209D]. Values were determined by incubating purified APT1[WT] and APT1[S209D] with increasing concentrations of DPP-3 substrates and measuring fluorescence for 240 s. The initial velocity of the reactions for APT1[WT] and APT1[S209D] activity was calculated by fitting the linear regression of the fluorescence vs. time. Results are averaged from six independent experiments.

| Substrate | Fluoresence/Time ($V_0$) | |
| --- | --- | --- |
| | APT1WT | APT1SD |
| 0.25µM | $0.2012 \pm 0.008627$ | $0.4704 \pm 0.01765$ |
| 0.5 µM | $0.2316 \pm 0.02656$ | $0.8083 \pm 0.03173$ |
| 1.0 µM | $0.5496 \pm 0.01478$ | $1.612 \pm 0.04416$ |
| 5.0 µM | $0.6573 \pm 0.03547$ | $4.627 \pm 0.2237$ |

DOI: https://doi.org/10.7554/eLife.34362.005

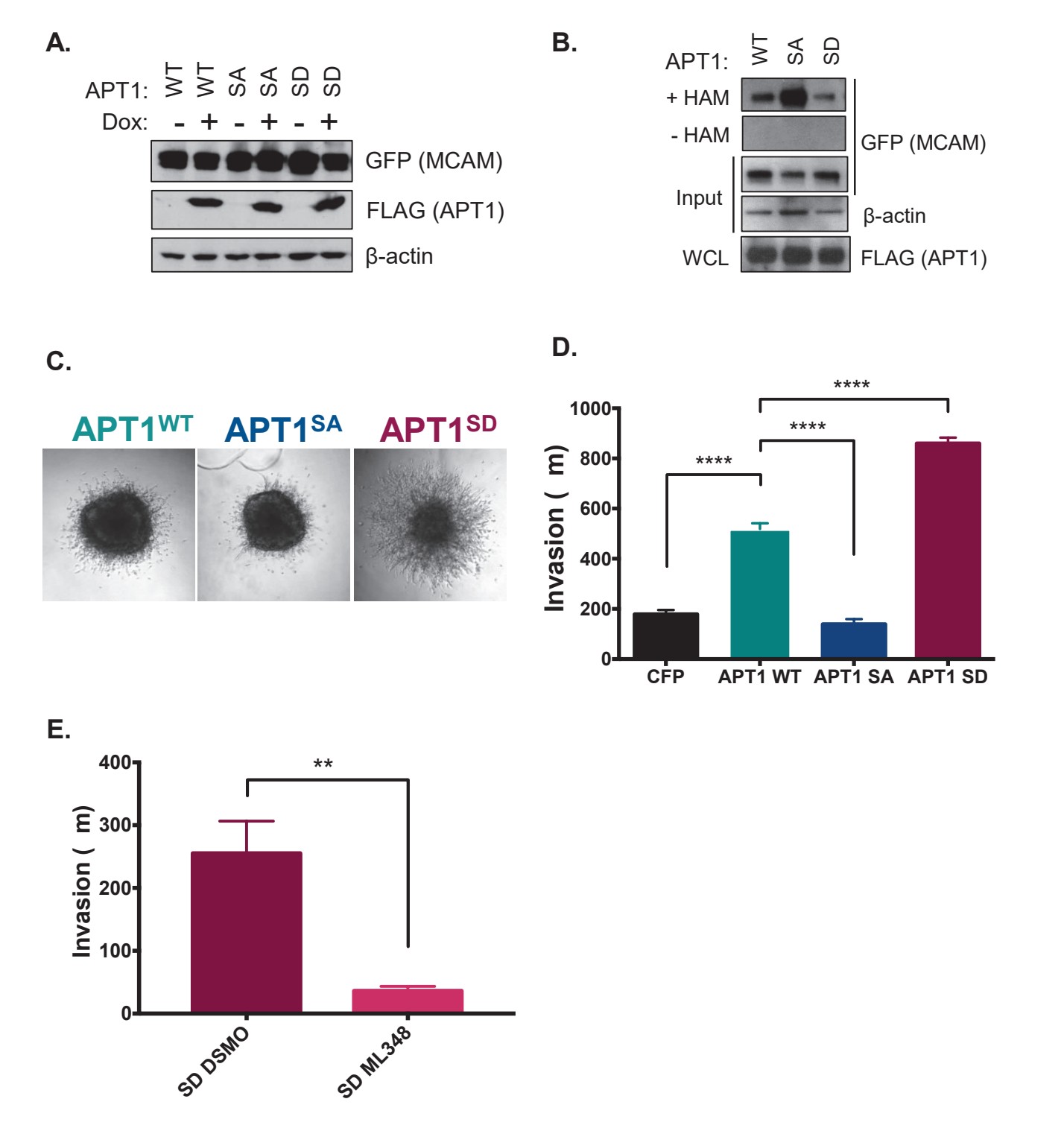

**Figure 3.** APT1 phosphorylation increases APT1 depalmitoylating activity in cells and increases melanoma invasion. (**A**) HEK 293T cells transfected with constitutively transcribed MCAM-GFP and doxycycline inducible APT1$^{WT}$-FLAG (WT), APT1$^{SA}$-FLAG (SA), or APT1$^{S209D}$-FLAG (SD). After 15 hr of induction with 1 µg/ml doxycycline, APT1 protein expression was determined by immunoblotting with anti-GFP (MCAM-GFP) and anti-FLAG (APT1-FLAG) antibodies. (**B**) An acyl biotin exchange (ABE) assay was used to measure MCAM palmitoylation in cell lysates from cell lines described in (**A**). In the ABE assay addition of hydroxylamine (+HAM) removes palmitate from cysteine residues that are then conjugated to biotin-HPDP. Biotinylated proteins are then isolated on streptavidin beads and palmitoylated proteins are analyzed by SDS-PAGE followed by immunoblotting. Hydroxylamine is

*Figure 3 continued on next page*

Figure 3 continued

withheld as a negative control (-HAM) MCAM-GFP is detected with anti-GFP antibodies and APT1-FLAG is detected with anti-FLAG antibodies. Results shown are a representative trial from three independent experiments. (C) WM239A melanoma cells expressing APT1$^{WT}$ (WT), APT1$^{SA}$ (SA), or APT1$^{S209D}$ (SD) were grown on agarose to form spheroids that were embedded in collagen and images were taken on day seven and the distance invaded was measured. Representative images from day seven are shown. (D) Quantification of WM239A spheroid invasion assay in (C). Error bars indicate s.e.m., n = 14–34 spheroids counted per condition, ****p<0.0001 by unpaired t-test analysis. Results shown are from four experiments. (E) Quantification of WM239A spheroid invasion assay of APT1$^{S209D}$ (SD) treated with DSMO control or 10 µM ML348, every other day for 7 days and distance invaded measured at day 7. Error bars indicate s.e.m., n = 6 spheroids counted per condition, **p=0.0017 by unpaired t-test analysis. Results shown are a representative trial from three independent experiments.

DOI: https://doi.org/10.7554/eLife.34362.006

## APT1 phosphorylation increases metastatic behavior in melanoma cells

Previous studies demonstrated increased APT1 expression results in increased invasion of melanoma cells embedded in collagen (*Wang et al., 2015*). To determine if this increased metastatic behavior was due to APT1 phosphorylation, we asked if APT1's phosphorylation state affects melanoma invasion using the APT1 phospho-mutants. Spheroids were formed from WM239A melanoma cells ectopically expressing CFP-FLAG-tagged APT1$^{WT}$, APT1$^{SA}$, or APT1$^{S209D}$, embedded in collagen, and the distance invaded was measured. We found that expression of APT1$^{SA}$ resulted in decreased invasion, similar to the negative control CFP, while APT1$^{S209D}$ significantly increased melanoma invasion (*Figure 3C and D*). Since these melanoma cells are not Wnt5a treated, this dramatic increase in APT1$^{S209D}$ melanoma invasion is due to the single point mutation that mimics phosphorylation alone. When the activity of APT1 is inhibited with the selective inhibitor ML348, invasion is blocked, demonstrating the increased invasion observed in cells expressing APT1$^{S209D}$ is mediated by APT1 activity (*Figure 3E*). Together these results indicate phosphorylation of Ser209 increases depalmitoylating activity in cells, in turn increasing metastatic cell behavior.

## Phosphorylation reduces APT1 dimerization

We next sought to determine how phosphorylation increases APT1 activity. We first asked if mutating the phospho-sites changed the thermostability of the protein using differential scanning fluorimetry of purified APT1$^{WT}$, APT1$^{SA}$, or APT1$^{S209D}$ mutants (*Figure 4A*). We found that each mutant possessed similar thermostability to that of APT1$^{WT}$, indicating that the point mutations are not destabilizing the folding of the protein (*Figure 4B*). We next asked if the existing crystal structure might provide insight into the mechanism of APT1 activity.

Previous crystal structures of APT1 revealed a weak asymmetric dimer in which the active site was occluded by the dimer interface, suggestive of an inactive dimer (*Devedjiev et al., 2000*; *Won et al., 2016*). Our interrogation of the crystal structure revealed that the phosphorylation sites 209 and 210 reside in the interface of the dimer and could potentially destabilize the dimeric form and decrease the inhibitory dimeric interaction. We measured the distance between the serine residues to the methionine (Met 65) residue on the opposite dimer and found a short distance of less than 4 Å (*Figure 4C*). We therefore hypothesized that phosphorylation of one of these sites would be less favorable for dimer formation. Studies confirming the dimeric state of APT1 in solution are lacking. We therefore sought to distinguish the APT1 monomer from the dimer by size-exclusion chromatography. When APT1 was loaded on the column at a concentration of 0.1 mg/ml both APT1$^{WT}$ and APT1$^{S209D}$ eluted at the same time as a 29kD standard. When the protein concentration is increased to 0.25 mg/ml a second peak of APT1$^{WT}$ begins to resolve closer to the 44kD standard (*Figure 4D*). When the APT1$^{WT}$ protein concentration was increased to 0.375 mg/ml a second peak was resolved near the 44kD standard that was not observed with APT1$^{S209D}$. Additionally, the peak at 29kD for APT1$^{WT}$ is abolished at this concentration (*Figure 4D*). The identity of the protein in the shifted peak was was confirmed as APT1 by visualizing the 25kD APT1 band by SDS-PAGE followed by Coomassie staining (*Figure 4D*). These results support the findings that APT1 dimerizes and phosphorylation of APT1 reduces dimerization.

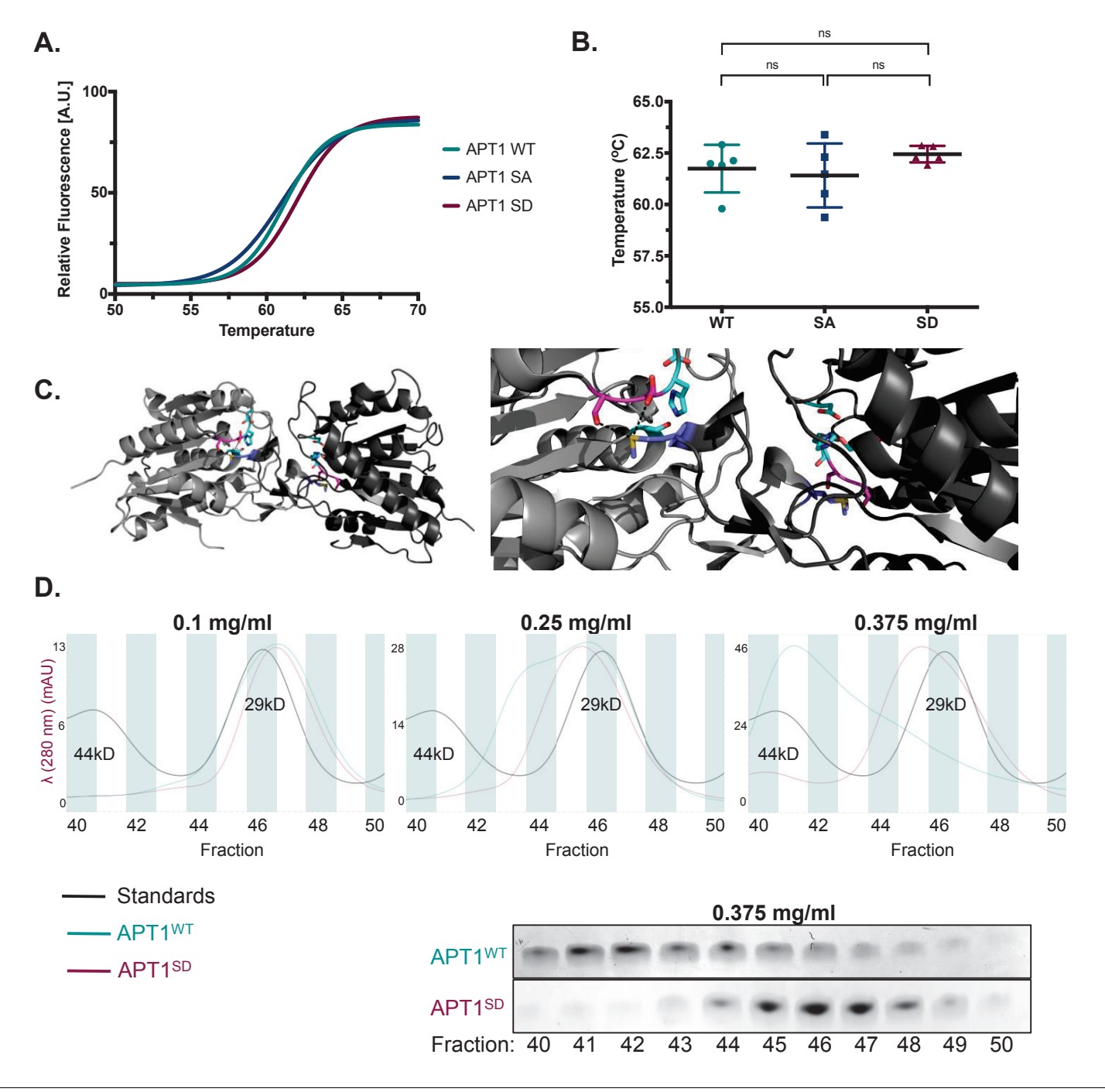

**Figure 4.** APT1 phosphorylation impedes APT1 dimer formation. (**A**) Differential scanning fluorimetry (DSF) of purified APT1[WT], APT1[SA] and APT1[S209D]. (**B**) Melting curves of purified APT1[WT], APT1[SA] and APT1[S209D] as determined by DSF. Results shown from DSF analysis are a representative trial from three independent experiments. (**C**) Left panel: Three-dimensional model of crystal structure of APT1 dimer interface. Ser209 and Ser210 (pink) are located 4.3 Å away from Met65 (blue) of the adjacent APT1 monomer. Catalytic triad Ser119, Asp174 and His208 are highlighted (cyan). Right panel: Zoom in of APT1 dimer interface. (**D**) Top panels: Chromatogram of purified APT1[WT] (cyan), APT1[S209D] (magenta), and molecular weight standards Ovalbumin (44kD) and Carbonic anhydrase (29kD) (black) from size-exclusion chromatography at three protein concentrations (Y-axis indicates the mAU of the APT1[SD]). Bottom panel: SDS-PAGE followed by coomassie staining of eluted fractions after separation by size-exclusion chromatography of APT1[WT] and APT1[S209D].

DOI: https://doi.org/10.7554/eLife.34362.007

## Serine 210 is mutated in cancer and increases APT1 activity

The APT1 gene is mutated in multiple tumor types, with the highest frequency of serine 210 mutated specifically to a leucine (*Figure 5A*) (*Gao et al., 2013* and *Cerami et al., 2012*). To investigate if this mutation also increases APT1 activity, we mutated serine 210 to leucine of APT1 (APT1$^{S210L}$) and measured its depalmitoylating activity by incubating purified protein with the DPP-3 probe. After measuring fluorescence over time, we observed an increase in depalmitoylating activity of APT1$^{S210L}$ compared to APT1$^{WT}$. When we compare the depalmitoylating activity of APT1$^{S210L}$ to APT1$^{S209D}$, we find that S210L exhibits an almost identical increase in activity to the phospho-mimetic APT1$^{S209D}$ (*Figure 5B*). We also observe similar inhibition of APT1 depalmitoylation by ML348 using the DPP-3 probe as a substrate for APT1$^{S209D}$ and APT1$^{S210L}$ (*Figure 5—figure supplement 1A*). These results demonstrate that the high activity of the APT1 phospho-mutants are still inhibited with ML348. To visualize how a S210L mutation would affect dimerization, we modeled the substitution on the three-dimensional crystal structure and determined a leucine at position 210 is located 5.5 Å from the methionine of the adjacent APT1 monomer (*Figure 5C*). With a neutral charge leucine positioned a small distance from the dimer interface, we propose steric hindrance disrupts the dimerization of APT1.

We next asked if this mutation would also enhance the ability of APT1 to promote cell invasion, similar to what we observe in APT1$^{S209D}$. To test the APT1$^{S210L}$ mutant's effect on cell behavior, we generated spheroids from WM239A melanoma cell expressing CFP-FLAG-tagged APT1$^{WT}$ and APT1$^{S210L}$, embedded them in collagen, and measured cell invasion each day over the course of 9 days. Expression of APT1$^{S210L}$ increased melanoma invasion when compared to APT1$^{WT}$ and negative control CFP (*Figure 5D*, *Figure 5—figure supplement 1B*). Similar to the phospho-mimetic APT1$^{S209D}$, treating spheroids expressing APT1$^{S210L}$ with ML348 significantly reduces cell invasion (*Figure 5—figure supplement 1C*). These results demonstrate that APT1$^{S210L}$ increases metastatic behavior compared to wild type and this is dependent on the catalytic activity of APT1.

In patients, increased Wnt5a expression is known to correlate with increased tumor grade and metastasis. To investigate if phospho-APT1 levels were also elevated in human melanoma samples and to establish human disease relevance, we stained human melanoma tumor arrays with our phospho-antibody (anti-pS209-APT1) and for total APT1 (anti-APT1 antibody). We discovered a correlation between high pAPT1 staining and increased tumor grade (*Figure 5E*). When looking at melanoma metastatic samples we observed the same correlation (*Figure 5F*, *Figure 5—figure supplement 1D*), implicating increased APT1 phosphorylation with increased tumor progression and metastasis.

## Discussion

Protein palmitoylation is often considered a constitutive modification required for correct protein localization and function. Here, we uncover a molecular switch that promotes protein depalmitoylation in response to non-canonical Wnt signaling. We demonstrate Wnt5a signaling induces APT1 phosphorylation and this activates APT1, increasing its depalmitoylating activity and thus depalmitoylation of its substrates. While phosphorylation of these serine residues has been observed in mass spectrometry studies, this is the first example of differential phosphorylation of APT1 in response to an extracellular signal. Ultimately, activation of APT1 signaling results in increased depalmitoylating activity, leading to increased melanoma invasion, and correlating with increased tumor grade and metastasis.

Understanding APT1 protein function has been hindered by a deficiency in assays for measuring thioesterase activity either in vitro or in cells. Here, we take advantage of a recently developed small molecule fluorophore, DPP-3, which allowed us to measure the activity of APT1 both in vitro and in cultured cells. This assay revealed the phospho-mimetic mutant APT1 has greater thioesterase activity compared to wild type APT1. The increase in activity was also observed by live cell imaging and the phospho-dead mutant had similar activity to wild type APT1. Using live imaging, we were able to demonstrate that the ability to increase endogenous APT1 activity is specific to Wnt5a and the noncanonical Wnt pathway, as we found Wnt3a lacks this ability.

The first crystal structure of APT1 revealed a dimer with the active site buried in the dimer interface (Devedjiev et al., 2000), suggesting the dimer must dissociate prior to interaction with its substrate. The phosphorylation sites are located on the edge of the hydrophobic channel distant from

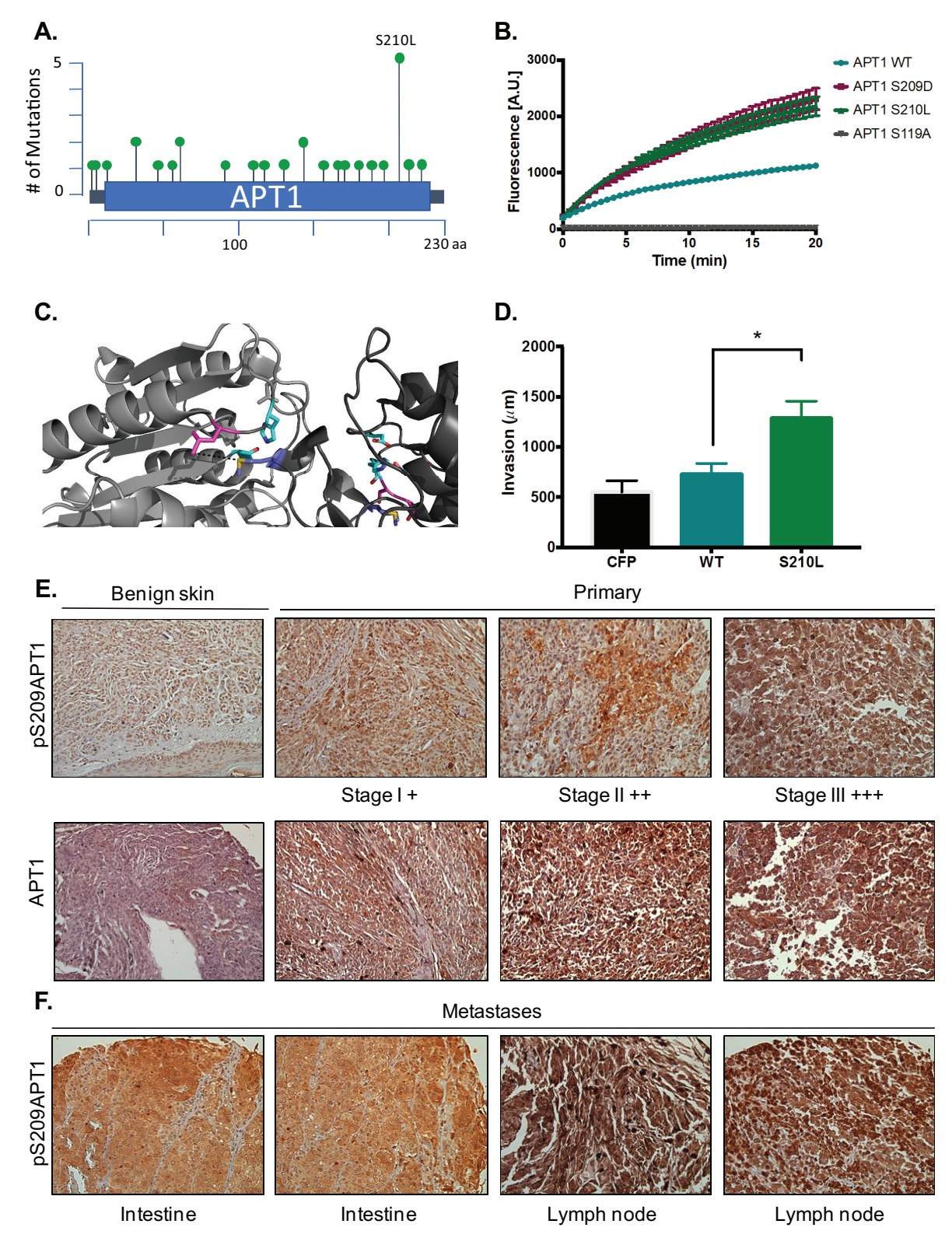

**Figure 5.** Increased phospho-APT1 staining correlates with increased melanoma tumor grade and metastasis. (A) Schematic of APT1 mutations from human tumor samples. Adapted from results generated using cBioPortal (*Gao et al., 2013* and *Cerami et al., 2012*). (B) In vitro fluorescence assays of 5 μM of DPP-3 with either 50 nM purified APT1$^{WT}$, APT1$^{S210L}$ APT1$^{S209D}$, or catalytically inactive APT1$^{S119A}$ and fluorescence emission was measured ($\lambda_{ex}$490/9 nm; $\lambda_{em}$545/20 nm). Error bars indicate s.e.m., n = 3 replicates per condition. Results shown are a representative trial from three independent
*Figure 5 continued on next page*

*Figure 5 continued*

experiments. (C) Three-dimensional model of crystal structure of APT1 dimer interface with Ser210 to Leu mutation. Ser210Leu (pink) is located 5.5 Å away from Met65 (blue) of the adjacent APT1 monomer. Catalytic triad Ser119, Asp174 and His208 are shown in cyan. (D) Quantification of WM239A CFP-FLAG (CFP), APT1$^{WT}$-CFP-FLAG (WT), and APT1$^{S210L}$-CFP-FLAG (S210L) spheroids embedded in collagen and distance invaded measured at day 8. Error bars indicate s.e.m., n = 8 spheroids per condition, *p=0.0150 by unpaired t-test analysis. Results shown are from three independent experiments. (E) Immunohistochemistry staining of melanoma tumors in a human melanoma tumor array using pS209-APT1 and APT1 antibodies. (F) Immunohistochemistry staining of metastatic tumor samples in a human melanoma tumor array using anti-pS209-APT1.

DOI: https://doi.org/10.7554/eLife.34362.008

The following figure supplement is available for figure 5:

**Figure supplement 1.** Response of the APT1 S210L mutant to the APT1 inhibitor ML348.

DOI: https://doi.org/10.7554/eLife.34362.009

the site of catalysis, but at the dimer interface. Our proposed model is phosphorylation impedes formation of an inactive dimer. Few studies have examined if APT1 forms a dimer in solution. Using size-exclusion chromatography we detected a complete shift of the APT1$^{WT}$ monomer population to the dimer. The shift was dependent on the concentration of APT1 since at the APT1$^{WT}$ dimer concentration APT1$^{S209D}$ remains a monomer. Furthermore, using three-dimensional modeling, we were able to determine the distance between the serines that are phosphorylated and the neighboring APT1 in the dimer interface to be 4.3 Å. With such a short distance between the serine and the adjacent APT1, it is consistent with the idea that phosphorylation of this serine would impede the dimerization of APT1. However, it is still unclear if this reduction in dimer formation is sufficient to account for the increase in APT1 activity. We still don't know if the higher activity of APT1$^{S209D}$ compared to APT1$^{WT}$ is caused by reduced dimer formation or a conformational change in the protein that increases activity. Alternatively, phosphorylation could alter the conformation of APT1 allowing it to accommodate the lipid substrate or increase the rate of catalysis independent of dimer formation.

When analyzing the results of the in vitro fluorogenic palmitoylation assay using the DPP-3 probe and the acyl biotin exchange assay measuring palmitoylation of MCAM as a protein substrate, there is a small discrepancy between the two assays for phospho-deficient mutant APT1$^{SA}$. In the acyl biotin exchange assay, we observe APT1$^{SA}$ displaying decreased activity compared to APT1$^{WT}$ cells. Whereas in our in vitro depalmitoylation assay with the DPP-3 probe, we do not observe a significant difference between the APT1$^{SA}$ mutant and APT1$^{WT}$. It is important to note that in whole cells there are other factors that play a role in this signaling pathway and this might contribute to the difference in the acyl biotin exchange assay, where we are measuring depalmitoylation of a protein APT1 substrate in this signaling pathway. It is possible that modifications of the substrate MCAM affect how its accessed or preference for depalmitoylation. Adaptor proteins may hinder or ease the depalmitoylation of the protein. Due to decreased melanoma invasion in collagen observed in APT1$^{SA}$ expressing cells, we hypothesize that the decreased depalmitoylation of MCAM may play a role in the APT1 mediated invasion. Thus, this is a more physiologically relevant measure of the depalmitoylation activity than solely using the artificial probe in vitro. Concerning the DPP-3 probe, it is still a reliable tool to assess APT1 depalmitoylation, when taken together with other data, which we have been able to do in this study.

The APT1 gene, LYPLA1, is amplified across a wide variety of tumor types, including 31.8%, 23.2% and 13.8%, in neuroendocrine prostate cancer, uterine carcinosarcoma and invasive breast carcinoma, respectively. Other cancers with amplified LYPLA1 include metastatic prostate adenocarcinoma, ovarian serous cystadenocarcinoma, and uveal melanoma (*Beltran et al., 2016*; *Cancer Genome Atlas Research Network et al., 2013*, and *Eirew et al., 2015*). Serine 210 is mutated not only in melanoma (*Krauthammer et al., 2012*), but also in multiple tumor types including colorectal adenocarcinoma and lung squamous cell carcinoma (*Giannakis et al., 2016*; *Cancer Genome Atlas Research Network, 2012*; *Campbell et al., 2016*). This bulky residue behaves similarly to the phosphorylated APT1 in vitro, suggesting this disease relevant mutation might result in increased depalmitoylation and thus increased invasion in patients. Using three-dimensional modeling, we find the leucine to be positioned 5.5 Å next the dimer interface and based on modeling, this mutation will not accommodate dimeric conformation. Due to this finding, this signaling pathway becomes increasingly important, not only in Wnt5a driven cancers mentioned above, but also for patients who present with APT1 mutations in other tumor types. There is a

possibility APT1 mutation in this phospho-site could be regulating cancer metastasis across other tumor types. Not surprisingly, the S210L mutation in APT1 is observed at a much lower frequency compared to well-established oncogenes frequently mutated in melanoma, such as BRAF or NRAS. We therefore think it is unlikely APT1 mutations contribute significantly to oncogenesis and tumor initiation. We did map the positions in the three-dimensional structure of the other mutations in APT1 catalogued in the TCGA database (*Figure 5A*). The only mutation that is positioned within the dimer interface is S210L, making it unlikely that the other low frequency mutations would increase APT1 activity through reduced dimerization. Therefore, APT1$^{S210L}$ may provide some selective advantage to tumor growth in vivo since it occurs at a higher frequency than the other mutations in APT1.

The data we present offers insights to the Wnt5a signaling pathway and its contribution to tumor progression and metastasis. Patients who present with elevated Wnt5a expression are in dire need of effective therapy and directly targeting the Wnt5a pathway is one possible avenue (*Prasad et al., 2015*). By identifying increased APT1 activity through either gene amplification or mutation of serine 210 as a driver of this pathway, we could expand the number of patients who can benefit from inhibition of the Wnt5a-APT1 pathway. This suggests APT1 signaling may play a larger role as a possible target for treating metastatic melanoma patients, especially those with APT1 amplification/mutations or disease recurrence after other treatments.

Here we show regulation of protein depalmitoylation by activation of the Wnt5a signaling pathway. This novel discovery allows us to better understand how secreted signals regulate protein function and polarized cell behavior. Wnt5a signaling directly regulates APT1 depalmitoylating activity and protein function, with activation or mutations in this pathway contributing to metastatic behavior. Novel therapeutics for Wnt5a driven cancers are lacking in the clinic. Understanding the signaling mechanism behind Wnt5a driven tumor progression and metastasis is imperative if we aim to create novel therapies against this pathway and to achieve our goal of increasing cancer patient survival.

## Materials and methods

### Plasmids

The coding sequence of APT1 was amplified from pcDNA3.1-APT1-FLAG and inserted into the AgeI/SalI sites of the pRRLSIN.cPPT.PGK-GFP.WPRE backbone. The APT1 S119A, S209D S209,210A and S210L mutations in were made by using the QuikChange site-directed mutagenesis kit (Agilent Technologies, Stratagene, Santa Clara, California) according to the manufacturer's protocol. APT1 wild type, S209D, S209,210A mutants were cloned into the AgeI/MluI sites of the inducible TRIPZ vector (Dharmacon). For bacterial expression APT1 wild type and mutants were cloned into the BamHI/XhoI sites of the pET-28a (+) plasmid.

### Purification of recombinant Wnt5a and treatments with Wnt5a and Wnt3a

Wnt5a was purified from mouse L-cells overexpressing human Wnt5a as described previously (*Willert et al., 2003*). Control cells were treated with control buffer with identical detergent conditions as the Wnt5a purification buffer. For experiments utilizing Wnt3a treatment, cells were treated with recombinant Wnt3a (R and D Systems, Minneapolis, MN) for 1 hr at 37°C before lysate was harvested as described.

### Cell lines and culture conditions

Metastatic melanoma cell line WM239A (BRAF V600D) (Wistar Institute) were cultured in RPMI 1640 medium (Gibco-BRL, Grand Island, NY) supplemented with 10% fetal bovine serum (FBS) (GE Life Sciences). Cell lines were authenticated routinely by short tandem repeat profiling by the Wistar Institute prior to use.

### Mass spectrometry analysis of APT1 phosphorylation sites

All chemicals used for preparation of mass spectrometry samples were of at least sequencing grade and purchased from Sigma-Aldrich (St Louis, MO), unless otherwise stated. The 1% TritonX-100

detergent was removed from samples prior MS analysis by precipitation using chloroform (CHCl3)-methanol (MeOH) precipitation [Wessel et al.]. The protein pellet from CHCl3-MeOH precipitation was resuspended in 6 M urea/2 M thiourea in 50 mM ammonium bicarbonate, pH 8.3 supplemented with Phosphatase and Protease Inhibitors Mix (Thermo Fisher Scientific, Waltham, MA). Samples were reduced with 10 mM DTT for 1 hr at room temperature and the carbamidomethylated with 20 mM iodoacetamide (IAA) for 30 min at room temperature in the dark. After alkylation proteins were digested first with endopeptidase Lys-C (Wako, Cambridge, MA; MS grade) for 3 hr, after which the solution was diluted 10 times with 20 mM ammonium bicarbonate, pH 8.3. Subsequently, samples were digested with trypsin (Promega, Madison, WI) at an enzyme to substrate ratio of approximately 1:50 for 12 hr at room temperature. After digestion, the samples were concentrated to the volume of ~100 µl by lyophilization. Phosphopeptide enrichment using titanium dioxide (TiO2) chromatographic resin was performed as previously described [Thingholm et al., Enghold et al.]. The lyophilized phosphorylated peptide samples were reconstituted in 0.1% trifluoroacetic acid (TFA) and desalted using Poros Oligo R3 RP (PerSeptive Biosystems, Framingham, MA) P200 columns. The peptide samples were subsequently lyophilized and stored at −80°C for further analysis.

Dried samples were resuspended in buffer-A (0.1% formic acid) and loaded onto an Easy-nLC system (Thermo Fisher Scientific, San Jose, CA), coupled online with an Orbitrap Fusion Tribrid mass spectrometer (Thermo Fisher Scientific, San Jose, CA). Peptides were loaded into a picofrit 25 cm long fused silica capillary column (75 µm inner diameter) packed in-house with reversed-phase Repro-Sil Pur C18-AQ 3 µm resin. The gradient length was 75 min. The gradient was from 2–26% buffer-B (100% ACN/0.1% formic acid) at a flow rate of 300 nl/min. The MS method was set up in a data-dependent acquisition (DDA) mode. For full MS scan, the mass range of 350–1200 m/z was analyzed in the Orbitrap at 120,000 FWHM (200 m/z) resolution and $5 \times 10e5$ AGC target value. HCD (higher energy collision dissociation) collision energy was set to 32, AGC target to 10e4 and maximum injection time to 200 msec. Detection of MS/MS fragment ions was performed in the ion trap in the rapid mode using the TopSpeed mode (2 s).

Raw MS-files were analyzed using Proteome Discoverer (v2.1, Thermo Scientific, Bremen, Germany). MS/MS spectra were converted to mgf files and searched against the UniProt-Human LYPLA1 (APT1) database (version June 2017) using SequestHT. Database searching was performed with the following parameters: precursor mass tolerance 10 ppm; MS/MS mass tolerance 0.6 Da; enzyme trypsin (Promega), with two missed cleavages allowed; fixed modification was cysteine carbamidomethylation; variable modifications were methionine oxidation, serine/threonine/tyrosine phosphorylation, asparagine and glutamine deamidation. Peptides were filtered for <1% false discovery rate, Sequest ion score >0.9. All MS-APT1 raw files have been deposited in the CHORUS database (maintained by the CHORUS project; https://chorusproject.org/) under project number 1456 (https://chorusproject.org/pages/dashboard.html#/projects/all?q=1456). Access to the data requires creating a free-account.

## Structural modeling

Modeling of catalytic triad (serine 119, aspartate 174, and histidine 208) and serine residues 209 and 210 identified to be phosphorylated by MS analysis was performed in MacPyMOL with the three-dimensional co-crystal structure of human APT1 in complex with an isoform selective inhibitor, ML348 at 1.55 A° (PDB 5SYM). Modeling of the distance between the distance between the oxygen side chain of the serine residues 209 and 210 in monomer A and the thiol group of methionine 65 in monomer B was performed in MacPyMOL with the three-dimensional co-crystal structure of human APT1 in complex with an isoform selective inhibitor, ML348 at 1.55 A° (PDB 5SYM). Modeling of serine 210 to leucine mutation identified in several cancers and the distance between the leucine backbone in monomer A and the thiol group of methionine 65 in monomer B was performed in MacPyMOL with the three-dimensional co-crystal structure of human APT1 in complex with an isoform selective inhibitor, ML348 at 1.55 A° (PDB 5SYM).

## Immunoprecipitation

WM239A cells ectopically expressing APT1-CFP-FLAG mutants were treated with control buffer or 150 ng/ml Wnt5a for specified time period and then lysed in lysis buffer containing 1% Triton-X 100, 50 mM Tris pH 7.5, 150 mM NaCl supplemented with protease and phosphatase inhibitors (1 µg/ml

leupeptin, 1 µg/ml aprotinin, 2 µg/ml pepstatin A, 1 mM PPi, 2 nM NaVO$_4$, 150 mM NaF). Insoluble cell debris was removed by centrifugation (13,000 RPM for 10 min at 4°C. Lysate was incubated with FLAG M2 magnetic beads (Sigma-Aldrich, St. Louis, MO) for 1.5 hr. Beads were washed with lysis buffer and protein was eluted using FLAG peptide for 1 hr at room temperature. Samples were separated by SDS-PAGE and transferred to either nitrocellulose membrane (Life Technologies, Thermo Fisher Scientific, Waltham, MA) for phospho-APT1 antibody (YenZym Antibodies, San Francisco, CA) or PVDF membrane (Millipore, Burlington, MA) for all other antibodies used.

## Western blot analysis and antibodies

Cells were harvested and lysed in 1% Triton X-100, 50 mM Tris pH 7.5, 150 mM NaCl supplemented with protease and phosphatase inhibitors (1 µg/ml leupeptin, 1 µg/ml aprotinin, 2 µg/ml pepstatin A, 1 mM PPi, 2 nM NaVO$_4$, 150 mM NaF). Insoluble cell debris was removed by centrifugation (13,000 RPM for 10 min at 4°C. The protein concentration was determined by DC Protein method (BioRad, Hercules, CA). Equal amounts of total protein were separated by SDS-PAGE and transferred to either nitrocellulose membrane (Life Technologies, Thermo Fisher Scientific, Waltham, MA) for phosphor-APT1 antibody or PVDF membrane (Millipore, Burlington, MA) for all other antibodies used. The nitrocellulose membranes were blocked with 5% bovine serum albumin (BSA) in TBST (TBS, 0.1% Tween). The PVDF membranes were blocked with 5% dry milk in TBST. All membranes were immunoblotted with different antibodies diluted in 5% BSA in TBST.

The rabbit anti-human APT1 antibody (Abcam, Cambridge, MA) was used at 1:1000. The rabbit anti-human beta-actin (Cell Signaling Technologies, Danvers, MA) was used at 1:5000. The mouse anti-humam MCAM antibody (Santa Cruz Biotechnology, Dallas, TX) was used at 1:1000. The secondary antibodies were HRP-conjugated 1:10000 diluted in 5% BSA in TBST. Membranes were washed three times with TBST between the different steps.

## Spheroid assay

96-well plates were coated with 50 µl per well of sterile 1.5% noble agar and solidified at room temperature for 10 min. 200 µl of 2.5 × 10$^4$ cells/ml cell suspension was added to each well. Spheroids formed at 37°C and at 4% CO$_2$ for 48 hr. Collagen matrices were prepared on ice using Pur Col purified bovine collagen (Advanced Biomatrix, Carlsbad, CA), Hyclone RPMI 1640 (5X) with sodium bicarbonate diluted to 1X in total volume and 10% FBS. Sterile NaOH was added to correct the collagen pH. 75 µl of collagen matrix was added to new wells and allowed to solidify at 37°C for 1 hr. Spheroids were resuspended in 125 µl of collagen matrix and transferred to wells containing 75 µl of collagen. After collagen solidified at 37°C, 100 µl of fresh medium was added on top of the collagen. Medium was changed every other day. Images were taken every 24 hr for 1–10 days. For the invasion assays including ML348 and LGK-974: 10 µM of ML348 (Sigma-Aldrich, St. Louis, MO) was added to fresh media on the spheroids and changed every other day.

## Acyl biotin exchange assay

The ABE assay was performed as described (*Wan et al., 2007*).

## Purification of APT1

Wild type and APT1 mutants (FLAG tagged) were cloned into the pET-28 plasmid backbone. Bacteria were grown overnight at 37°C in the presence of 100 µg/ml ampicillin and 50 µg/ml chloramphenicol. Next, fresh LB or TB was inoculated with the overnight bacteria and grown for 4 hr at 37°C until OD$_{600}$ = 0.6–1.0. Expression of APT1 was induced with final concentration of 1 mM IPTG for 2 hr at 37°C or overnight at 18°C. Cells were pelleted at 8000 RPM for 15 min at 4°C. Pellet was resuspended in 50 mM HEPES pH 8.0, 300 mM NaCl, 1% Triton X-100, 20 mM Imidazole (Sigma), 1 mM Phenylmethylsulfonyl fluoride (PMSF), 1 mM dithiothreitol or 25 mM HEPES pH 7.5, 500 mM NaCl, 10% glycerol, 1 mM PMSF, 10 mM 2-Mercaptoethanol, complete EDTA free protease inhibitor cocktail (Pierce). Lysate was sonicated at 50% duty cycle for 30 s pulses two times on ice or 50% duty cycle for 30 pulses on and 30 pulses off for 30 min at 4°C. Lysate was centrifuged at 12,000 RPM for 15 min at 4°C. Supernatant was incubated with Ni Sepharose 6 Fast Flow beads (GE Healthcare) at 4°C rocking for 3 hr or overnight. Beads were washed three times with wash buffer (50 mM HEPES pH 8.0, 300 mM NaCl, 40 mM Imidazole) or 500 mL of wash buffer (25 mM HEPES pH 7.5, 500 mM

NaCl, 10% glycerol, 40 mM Imidazole, 10 mM 2-Mercaptoethanol). Protein was eluted using 50 mM HEPES pH 8.0, 300 mM NaCl, 250 mM Imidazole by rocking for 45 min at 4°C or 50 mL of 25 mM HEPES pH 7.5, 500 mM NaCl, 300 mM Imidazole, 10 mM 2-Mercaptoethanol at 4°C. For SEC experiments, 50 mL of eluted APT1 was dialyzed against 25 mM HEPES pH7.5, 150 mM NaCl, 10 mM 2-Mercaptoethanol overnight at 4°C. Dialyzed APT1 was concentrated using a Amicon Ultra-15 centrifugal filter unit to 500 μL and injected onto NGC Liquid Chromotography System (Bio-Rad) equipped with a Superdex 200 Increase 10/300 GL (GE) size-exclusion chromatography (SEC) equilibrated in 25 mM HEPES pH7.5, 150 mM NaCl, 10 mM 2-Mercaptoethanol. Fractions containing homogeneous APT1 were utilized for analytical SEC at various protein concentrations.

## Fluorogenic palmitoyl thioesterase assays

APT1 was assayed for depalmitoylating activity as described previously (*Kathayat et al., 2017*).

## Fluorogenic palmitoyl thioesterase assay from immunopurified APT1-CFP-FLAG

WM239A cells ectopically expressing APT1-CFP-FLAG mutants were treated with control buffer or 150 ng/ml of purified Wnt5a and lysed in lysis buffer containing 1% Triton-X 100, 50 mM Tris pH 7.5, 150 mM NaCl supplemented with protease and phosphatase inhibitors (1 μg/ml leupeptin, 1 μg/ml aprotinin, 2 μg/ml pepstatin A, 1 mM PPi, 2 nM NaVO$_4$, 150 mM NaF). Insoluble cell debris was removed by centrifugation (13,000 RPM for 10 min at 4°C. Lysate was incubated with FLAG M2 magnetic beads (Sigma-Aldrich, St. Louis, MO) for 1.5 hr. Beads were washed with lysis buffer and fluorogenic palmitoyl thioesterase assay was performed as described previously (*Kathayat et al., 2017*).

## Differential scanning fluorimetry

Differential scanning fluorimetry data were collected on a QuantStudio3TM Real-Time PCR Detection System (Applied Biosystems) at a APT1 protein concentration of 1.42 μM in either 50 mM HEPES pH 8.0 using SYPRO orange as described previously (*Niesen et al., 2007*). The melting temperature was calculated by fitting the normalized data curve to the Boltzmann sigmoid equation in Prism 6 (GraphPad) as described previously (*Niesen et al., 2007*).

## Size-exclusion chromatography

Homogenous APT1$^{WT}$ or APT1$^{S209D}$ at 0.1 mg/mL, 0.25 mg/mL, or 0.375 mg/mL were injected onto NGC Liquid Chromotography System (Bio-Rad) equipped with a Superdex 200 Increase 10/300 GL (GE) size-exclusion chromotography (SEC) equilibrated in 25 mM HEPES pH 7.5, 150 mM NaCl, 10 mM 2-Mercaptoethanol. Fractions corresponding to molecular weight standards between 44kD (Ovalbumin) and 29kD (Carbonic anhydrase) were evaluated by Coomassie Brilliant Blue (CBB).

## Enzyme kinetics

Purified APT1 protein (purification mentioned above) was incubated with increasing concentrations of DPP-3 substrate and fluorescence was measured over time to measure depalmitoylation (*Kathayat et al., 2017*). Initial velocities were calculated by fitting the linear regression of the fluorescence vs. time for APT1$^{WT}$ or APT1$^{S209D}$ at each DPP-3 substrate concentration.

## Immunohistochemistry

Melanoma tumor arrays (US Biomax, Inc, ME1004b and ME1004e) were immunostained with primary antibody 1:50 anti-pS209APT1 and 1:200 anti-APT1. Staining procedure was performed as described in *Walter et al. (2017)*. Images were obtained using 20X objective.

## Statistics

\* denotes a P-value between 0.0150–0.0205, \*\* denotes a P-value of 0.0017 and \*\*\*\* denotes a P-value of less than 0.0001 in an unpaired, two-tailed Students t-test, assuming normal distribution and equal variance. Each experiment was performed at least three times.

## Acknowledgements

This work was supported by NIH Grants R01CA181633 and AI118891, ACS Grants RSG-15-027-01, IRG –78-002-34. This project is funded, in part, under a grant with the Pennsylvania Department of Health. The Department specifically disclaims responsibility for any analyses, interpretations or conclusions. D.C.B. is funded by the Pew Biomedical Scholars Program #29622. We would like to thank Rachel Ekaireb for initial characterization of the APT1 mutants. We would also like to thank Dr. Todd Ridky, Dr. Christopher Natale, and Dr. Luca Busino for their help and support throughout the research project.

## Additional information

### Competing interests

Rahul Singh Kathayat, Bryan C Dickinson: has applied for a provisional patent for DPP-3. A patent number has not been assigned. The other authors declare that no competing interests exist.

### Funding

| Funder | Grant reference number | Author |
|---|---|---|
| National Institute of Allergy and Infectious Diseases | AI118891 | Benjamin A Garcia |
| National Cancer Institute | CA181633 | Eric S Witze |
| American Cancer Society | RSG-15-027-01 | Eric S Witze |
| American Cancer Society | IRG –78-002-34 | Eric S Witze |

The funders had no role in study design, data collection and interpretation, or the decision to submit the work for publication.

### Author contributions

Rochelle Shirin Sadeghi, Conceptualization, Formal analysis, Supervision, Validation, Investigation, Visualization, Methodology, Writing—original draft, Project administration, Writing—review and editing; Katarzyna Kulej, Validation, Investigation, Visualization, Methodology; Rahul Singh Kathayat, Resources; Benjamin A Garcia, Resources, Supervision, Validation, Visualization, Writing—review and editing; Bryan C Dickinson, Conceptualization, Resources, Formal analysis, Validation, Investigation, Visualization, Methodology, Writing—original draft, Project administration, Writing—review and editing; Donita C Brady, Conceptualization, Resources, Formal analysis, Supervision, Validation, Visualization, Methodology; Eric S Witze, Conceptualization, Supervision, Funding acquisition, Methodology, Project administration, Writing—review and editing

### Author ORCIDs

Rahul Singh Kathayat (iD) http://orcid.org/0000-0002-9159-2413
Bryan C Dickinson (iD) http://orcid.org/0000-0002-9616-1911
Eric S Witze (iD) http://orcid.org/0000-0002-7699-4879

### Decision letter and Author response

Decision letter https://doi.org/10.7554/eLife.34362.016
Author response https://doi.org/10.7554/eLife.34362.017

## Additional files

### Supplementary files

• Transparent reporting form
DOI: https://doi.org/10.7554/eLife.34362.010

## Major datasets

The following dataset was generated:

| Author(s) | Year | Dataset title | Dataset URL | Database, license, and accessibility information |
|---|---|---|---|---|
| Sadeghi RS, Kulej K, Kathayat RS, Garcia BA, Dickinson BC, Brady DC, Witze E | 2018 | Wnt5a Signaling in melanoma progression and metastasis | https://chorusproject.org/pages/dashboard.html#/projects/all?q=1456 | Available at the CHORUS database (access to the data requires creating a free-account) |

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
