## [Decision Letter]

Thank you for submitting your article "Wnt5a Signaling Induced Phosphorylation Increases APT1 Activity and Promotes Melanoma Metastatic Behavior" for consideration by *eLife*. Your article has been favorably evaluated by Jonathan Cooper (Senior Editor) and three reviewers. The reviewers have opted to remain anonymous.

The reviewers have discussed the reviews with one another and the Reviewing Editor has drafted this letter to crystallize our thoughts on the work that would be necessary to move forward with the publication of your paper. At this point we ask that you consider the essential revisions below.

Summary:

The authors describe a novel pathway in which Wnt5a regulates cell adhesion and invasion of metastatic cells by triggering phosphorylation of the acyl protein thioesterase 1 (APT1), which enhances its activity; prior work showed that Wnt5a mediated invasion is reduced by inhibitors of APT proteins. Adhesion molecules including CD44 and MCAM are palmitoylated, and removing this modification increases migration and invasion. The authors demonstrate using mass spectrometry Wnt5a-induced phosphorylation of APT1 at two residues, one of which is mutated in melanoma. Using a phosphomimetic S209D mutant, they show that this modification enhances depalmitoylation activity, and when expressed in melanoma cells enhances invasiveness. Disrupting APT1 phosphorylation in response to Wnt5a (by site-directed mutagenesis of APT1) compromises the migratory capacity of cells overexpressing APT1. They conclude that phosphorylation-stimulated decreases in palmitoylation are accompanied by cell morphology changes consistent with models of metastasis. The study also introduces new antibodies for detecting phosphorylated APT1 that will likely be useful for futures studies focused this gene family and more generally the role of protein palmitoylation in the regulation of cancer.

Essential revisions:

The reviewers agree that the manuscript reports a potentially novel pathway for Wnt5a control of cell adhesion and invasion. However, there are a number of experiments needed to strengthen the conclusions of the paper. Also, there were particular concerns about the proposed mechanism in which phosphorylation of the S209-S210 region controls enzyme activity by disrupting inhibitory dimerization.

1) The study heavily relies on a single purified Wnt protein. There is no attempt to demonstrate specificity of this response using either a Wnt5a that has no activity (mutation of the lipidation site or use of an inhibitor of Wnt acylation, for example), or other purified Wnt proteins. Given that other types of stimulation (such as growth factors) can also influence APT1 activity, the potential contribution of protein contaminants (or even the buffer/detergents used to maintain Wnt5a solubility) to APT1 phosphorylation/response should be examined. A control buffer should be applied in parallel studies using recombinant Wnt5a if this hasn't already been done (not mentioned in Materials and methods section).

2) The role of cell autonomous Wnt signaling in regulating APT1-dependent tumor cell outgrowth (Figure 3C and Figure 5—figure supplement 1) should be evaluated pharmacologically or with a genetic approach to increase confidence in the relevance of this overexpression approach for reporting physiological cancer cell behavior.

3) Have the authors tested whether protein kinase inhibitors will prevent the Wnt5a-stimulated effects of APT1 activity? Likewise, have the authors assessed the capacity of phosphatase inhibitors in any of the assays? Similarly, the in vitro palmitoylation assays can be easily be validated for their capacity to response to phosphorylation by testing whether pretreatment with purified or enriched enzyme with a phosphatase (e.g. lambda phosphatase). Data in Figures 1 and 2 suggest that wt APT1 may display basal levels of phosphorylation. Phosphatase pretreatment should make the differences between the treated and untreated samples larger, including the mutants.

4) In Figure 2 the authors report the depalmitoylating capacity of APT1 using an interesting in vitro assay. Some attempt to express the data quantitatively should be made. Some kinetic constants such as initial rates and maximal velocities of depalmitoylation should be calculated and presented so that statistical comparisons may be drawn. This is quite important when considering the activity of Wnt5A-treated esterase activity.

5) Similarly, the small but apparently significant increase in esterase activity of ATP1 following Wnt5A treatment should also be directly compared with two SA and SD mutants. The increase in activity in response to Wnt5A may be due to other signaling pathways and not necessarily through the same mechanism proposed by the authors.

6) Several concerns regarding the experiments purporting to show that phosphorylation of these serines enhance activity by disrupting an inactive dimer. The dimer was seen in crystal structures and appears to occlude the active site.

a) The authors claim that the recombinant material used in their studies is a dimer by gel filtration, but no data are shown. Looking at the structure, the serines in the dimer interface would not be accessible to a kinase, so there must at least be a monomer-dimer equilibrium. They need to show the gel filtration data of WT and mutant APT1.

b) DSC crosslinking is used to suggest that there is a dimer, but the level of crosslinking is weak (Figure 4D). Moreover, DSC is a short, lysine specific crosslinker; inspection of the dimer structure suggests that no lysines are close enough for DSC crosslinking. Thus, it is not clear what they are seeing in this experiment.

c) Given the proximity of the serines to the active site, it seems equally plausible that their mutants or phosphorylation preclude or at least hinder access to the active site, or perhaps alter the structure to reduce catalysis. Could this also explain S210L?

d) Given the mutational data shown in Figure 5A, where do these mutations map? Are they consistent with the dimer hypothesis? What might they be doing?

7) The data presented in Figure 5B and described in the Results section are a little misleading. The data in Figure 5B for the wt and SD mutants appear to be the identical data presented in Figure 2A (exactly the same error bars). The experiment should be repeated under the identical conditions. Perhaps the assays on the SL mutant were conducted at the same time as the other mutants. If this is the case then the manuscript should reflect this or the figure should be modified to express this (e.g. presenting the wt and SD data as a 'dissolved' or slightly 'transparent' form).

8) Ser210 may be the most frequently mutated residue in APT1 but at 5 mutations out of xxx tumors, its potential significance should be discussed in the context of mutation frequencies found in well-established drivers of melanoma such as BRAF or NRAS for example.

[Editors' note: further revisions were requested prior to acceptance, as described below.]

Thank you for resubmitting your work entitled "Wnt5a Signaling Induced Phosphorylation Increases APT1 Activity and Promotes Melanoma Metastatic Behavior" for further consideration at *eLife*. Your revised article has been favorably evaluated by Jonathan Cooper (Senior Editor) and a Reviewing Editor.

The manuscript has been improved but there are some remaining issues that need to be addressed before acceptance, as outlined below:

The authors have strengthened the paper considerably with new experiments that support the model for the effect of Wnt5a on adhesion and invasiveness of tumor cells. These include comparison of the activity of Wnt3a with that of Wnt5a to bolster the argument that the effect is Wnt5a/non-canonical pathway specific; use of two general kinase inhibitors to further probe the key role of phosphorylation; demonstrating that inhibition of APT1 blocks the effect of the mutants; quantifying the effect of the S209D on enzyme activity; and presentation of data showing dimerization of the purified wt, but not mutant, protein.

1) In the subsection “APT1 Phosphorylation Increases Depalmitoylating Activity”, it might be useful to explicitly note that the identity of the kinase is unknown.

2) The dimerization mechanism model requires further clarification and discussion:

- The authors should be absolutely clear as to the basis of the statement "…phosphorylation of APT1 does not completely block dimerization…" Is it the very small rise at the highest concentration at 0.375 mg/ml SEC trace? The presence of "dimer" in the S209D gel in Figure 4D looks like an artifact of boiling in reducing sample buffer, which can cause some disulfide formation; in fact, C173 is surface exposed. Or is this inferred by the fact that the mutant retains some activity? If the latter, this is an interpretation based on the model and is something of a circular argument.

- The model for dimerization assumes the dimer interface seen in the asymmetric unit of the crystal, which as noted in the original review is not compatible with the DSC crosslinking. The explanation provided by the authors is problematic, as it would imply transient formation of different dimers. There is another dimer in crystal lattice that could account for observed crosslinking, e.g. between K219 and K90. Perhaps this dimer could limit access of substrate, and electrostatic repulsion between D101 might explain S209D; or, there are allosteric effects of S209 and S210 on formation of this "alternative" dimer interface (this also touches on the analysis of the other mutations shown in Figure 5). In any case, either explicit discussion of the DSC result is needed or these crosslinking data should be removed. Without a definitive experimental analysis of the true dimer interface, the connection between activity and dimer formation remains correlative, and the effect of the mutants at S209 and S210 may well be due to other effects as they now note.

- The traces in Figure 4E should be labeled with the apparent MW of the standard peaks. Also, the fraction numbers are unreadable; perhaps colored bars indicating which fractions have been run on the gel would be better.

---

## [Author Response]

Essential revisions:The reviewers agree that the manuscript reports a potentially novel pathway for Wnt5a control of cell adhesion and invasion. However, there are a number of experiments needed to strengthen the conclusions of the paper. Also, there were particular concerns about the proposed mechanism in which phosphorylation of the S209-S210 region controls enzyme activity by disrupting inhibitory dimerization.1) The study heavily relies on a single purified Wnt protein. There is no attempt to demonstrate specificity of this response using either a Wnt5a that has no activity (mutation of the lipidation site or use of an inhibitor of Wnt acylation, for example), or other purified Wnt proteins. Given that other types of stimulation (such as growth factors) can also influence APT1 activity, the potential contribution of protein contaminants (or even the buffer/detergents used to maintain Wnt5a solubility) to APT1 phosphorylation/response should be examined. A control buffer should be applied in parallel studies using recombinant Wnt5a if this hasn't already been done (not mentioned in Materials and methods section).

We tried to use Wnt acylation inhibitors, however the WM239A cells secrete very little Wnt5a relative to other melanoma cell lines and for this reason we think the acylation inhibitor LGK-974 had very little effect on APT1 function. Instead, to determine if the increase in APT1 activity is specific to non-canonical Wnt5a we repeated the live cell depalmitoylation assay using purified Wnt3a, a conical Wnt ligand. We first confirmed the cells in our assay could respond to purified Wnt3a and found that treating APT1^WT^ expressing WM239A cells with 75 and 150ng/ml of Wnt3a for 1 hour was sufficient to stabilize β-catenin, an established measure of canonical Wnt signaling activation (Figure 2—figure supplement 1C). We then measured APT1 activity in live WM239A APT1^WT^ cells treated with either control buffer, 150 ng/ml Wnt5a, or 150 ng/ml Wnt3a and found Wnt3a is unable to increase APT1 activity above levels observed with control buffer after 45 minutes (Figure 2E). In Author response image 1 we have included results from three time points for the reviewers. Each dot represents the fluorescence emission from a single cell quantified by measuring the pixel density of the fluorescent image. The image quantification is described in the figure legends. In the manuscript only the 30 minute time point is shown, here we show time points at 15, 30 (Figure 2E), and 45 minutes.

2) The role of cell autonomous Wnt signaling in regulating APT1-dependent tumor cell outgrowth (Figure 3C and Figure 5—figure supplement 1) should be evaluated pharmacologically or with a genetic approach to increase confidence in the relevance of this overexpression approach for reporting physiological cancer cell behavior.

We have pharmacologically evaluated the APT1-dependent tumor outgrowth by treating the APT1^S209D^ and APT1^S210L^ mutants with the APT1 inhibitor ML348. When the melanoma spheroid invasion assay is carried out in the presence of ML348 invasion caused by expressing the APT1^S209D^ or APT1^S210L^ mutant is blocked (Figure 3E and Figure 5—figure supplement 1C). These results confirm the increase in invasion caused by mutations in the APT1 phospho-sites are dependent on the activity of APT1.

3) Have the authors tested whether protein kinase inhibitors will prevent the Wnt5a-stimulated effects of APT1 activity? Likewise, have the authors assessed the capacity of phosphatase inhibitors in any of the assays? Similarly, the in vitro palmitoylation assays can be easily be validated for their capacity to response to phosphorylation by testing whether pretreatment with purified or enriched enzyme with a phosphatase (e.g. lambda phosphatase). Data in Figures 1 and 2 suggest that wt APT1 may display basal levels of phosphorylation. Phosphatase pretreatment should make the differences between the treated and untreated samples larger, including the mutants.

We tested two different ATP-competitive kinase inhibitors, a serine/threonine kinase inhibitor BI- D1870 that inhibits p90 RSK at an IC50 range of 15-30 nM and the pan-kinase inhibitor staurosporine, to test for their ability to inhibit Wnt5a induced APT1 activity in Wnt5a treated cells. WM239A cells expressing APT1^WT^ were treated with either DSMO control, 10 µM BI-D1870, or 0.2 µM staurosporine for 1 hour, loaded with 10 µM DPP-3, treated with Wnt5a and analyzed by live-cell fluorescence microscopy after 15, 30, and 45 minutes. Both kinase inhibitors block Wnt5a induced increased APT1 activity, suggesting the APT1 phosphorylation is the mechanism for increasing activity. We intentionally used high concentrations of inhibitors since the objective was to nonspecifically inhibit kinases since we have yet to identify the relevant kinase. We chose to include the 30 minute time point for Wnt5a treatment in the paper (Figure 2G), even though it worked at all time points.

**Author response image 2. respfig2:** 

We agree the phosphatase experiment is an excellent idea. However, when we attempted to inhibit the Wnt5a induced APT1 activity with lambda phosphatase treatment, it was extremely challenging to retain the activity of APT1 pulled down from cells. The procedure required an anti-FLAG pull-down followed by extensive washes then the lambda phosphatase reaction, and additional washes prior to the enzymatic assay. After all of the steps we had very low APT1 activity across all samples. We suspect this is due to the extra incubations in the lambda phosphatase buffers and extra washes that lengthened the duration of the experiments. We hope that showing the increase in APT1 activity induced by Wnt5a is inhibited with two kinase inhibitors in live cells is adequate to strengthen the conclusion that the phosphorylation of APT1 in response to Wnt5a signaling is the mechanism of increased APT1 activity.

4) In Figure 2 the authors report the depalmitoylating capacity of APT1 using an interesting in vitro assay. Some attempt to express the data quantitatively should be made. Some kinetic constants such as initial rates and maximal velocities of depalmitoylation should be calculated and presented so that statistical comparisons may be drawn. This is quite important when considering the activity of Wnt5A-treated esterase activity.

To determine the initial velocity rates for APT1^WT^ and APT1^S209D^ using the DPP-3 probe, we incubated purified APT1^WT^ and APT1^S209D^ with increasing concentrations of DPP-3 substrate. Initial velocities were calculated by fitting the linear regression of the fluorescence vs. time for APT1^WT^ or APT1^SD^ at each DPP-3 substrate concentration (Figure 2B, C). We measured the initial velocity of APT1^S209D^ to be between 2 to 4-fold higher than APT1WT. We have included Table 1 which presents the initial velocities and the standard error of the mean for each substrate concentration tested. We did not determine the maximal velocity for APT1^S209D^ because we were unable to reach saturation with our highest concentration of substrate. At concentrations higher than 5 µM we believe we had problems with solubility of the substrate. Nevertheless, the increased enzyme kinetics of phospho- mimetic APT1^S209D^ is quantitatively higher than APT1^WT^.

5) Similarly, the small but apparently significant increase in esterase activity of ATP1 following Wnt5A treatment should also be directly compared with two SA and SD mutants. The increase in activity in response to Wnt5A may be due to other signaling pathways and not necessarily through the same mechanism proposed by the authors.

We performed the DPP-3 assay using live-cell imaging on APT1^WT^, APT1^SA^, and APT1^SD^ mutants and the results are represented in Figure 2F. As expected, Wnt5a treatment increases depalmitoylating activity of APT1WT to levels similar to the APT1^S209D^ mutant. Wnt5a has no significant effect on APT1^SA^ or APT1^S209D^. The inability of Wnt5a to further increase the activity of APT1^S209D^ is important because this would suggest the Wnt5a mediated increase is entirely driven by APT1 phosphorylation.

6) Several concerns regarding the experiments purporting to show that phosphorylation of these serines enhance activity by disrupting an inactive dimer. The dimer was seen in crystal structures and appears to occlude the active site.a) The authors claim that the recombinant material used in their studies is a dimer by gel filtration, but no data are shown. Looking at the structure, the serines in the dimer interface would not be accessible to a kinase, so there must at least be a monomer-dimer equilibrium. They need to show the gel filtration data of WT and mutant APT1.

We have included the results of the gel filtration experiment after repeating the experiment using three different concentrations of APT1 (25 kD) protein in Figure 4E. When APT1 was loaded on the column at a concentration of 0.1 mg/ml both APT1^WT^ and APT1^S209D^ eluted at the same time as a 29kD standard. When the protein concentration is increased to 0.25 mg/ml a second peak of APT1^WT^ begins to resolve closer to the 44kD standard (Figure 4E). When the APT1^WT^ protein concentration was increased to 0.375 mg/ml a second peak was resolved near the 44kD standard that was not observed with APT1^S209D^. The peak at 29kD for APT1^WT^ is abolished at this concentration (Figure 4E). These results support the findings in the crosslinking experiments, indicating APT1 dimerizes and phosphorylation of APT1 does not completely block dimerization, but impedes dimer formation.

b) DSC crosslinking is used to suggest that there is a dimer, but the level of crosslinking is weak (Figure 4D). Moreover, DSC is a short, lysine specific crosslinker; inspection of the dimer structure suggests that no lysines are close enough for DSC crosslinking. Thus, it is not clear what they are seeing in this experiment.

It is possible the APT1 dimer is detected by chemical crosslinking without lysines within close proximity to each other in the active site because the interaction between monomers is more dynamic than what is represented in the crystal structure. The transient interactions that form when the protein monomers interact may bring lysine residues outside of the dimer interface within crosslinking distance and this intermediate and not the stable dimer is what is captured in this assay. We feel the SEC data support this hypothesis.

c) Given the proximity of the serines to the active site, it seems equally plausible that their mutants or phosphorylation preclude or at least hinder access to the active site, or perhaps alter the structure to reduce catalysis. Could this also explain S210L?

We agree that increased access to or altered structure of the active site caused by phosphorylation are alternative mechanisms to regulate catalysis. We now discuss these possibilities in the Discussion section.

“However, it is still unclear if this reduction in dimer formation is sufficient to account for the increase in APT1 activity. Alternatively, phosphorylation could alter the conformation of APT1 allowing it to accommodate the lipid substrate or increase the rate of catalysis.”

d) Given the mutational data shown in Figure 5A, where do these mutations map? Are they consistent with the dimer hypothesis? What might they be doing?

We mapped the positions of all of the mutations identified in the TCGA database (Figure 5A) on the three-dimensional crystal structure to determine if any localize to the dimer interface. The only mutation that maps in the dimer interface is S210L which is the only mutation to occur at a frequency higher than 2. Therefore, we think it is unlikely that the mutations other than S210L would disrupt the dimer. We think it is likely the low frequency mutations are passenger mutations that have minimal effect on APT1 activity or tumor growth or fitness. We now discuss this point in greater length in the Discussion.

“We did map the positions in the three-dimensional structure of the other mutations in APT1 catalogued in the TCGA database (Figure 5A). The only mutation that is positioned within the dimer interface is S210L, making it unlikely that the other low frequency mutations would increase APT1 activity through reduced dimerization. Therefore, APT1S210L may provide some selective advantage to tumor growth in vivo since it occurs at a higher frequency than the other mutations in APT1.”

7) The data presented in Figure 5B and described in the Results section are a little misleading. The data in Figure 5B for the wt and SD mutants appear to be the identical data presented in Figure 2A (exactly the same error bars). The experiment should be repeated under the identical conditions. Perhaps the assays on the SL mutant were conducted at the same time as the other mutants. If this is the case then the manuscript should reflect this or the figure should be modified to express this (e.g. presenting the wt and SD data as a 'dissolved' or slightly 'transparent' form).

We apologize that the figure was misleading, it was not our intent to misrepresent the data. To remove any confusion, we have repeated the experiment under identical conditions comparing purified APT1^WT^, APT1^S209D^, APT1^S210L^ and APT1^S119A^. We agree that this will remove any possible misconceptions about how the experiment was performed. Below is the representation of an experiment including all mutants analyzed at the same time shown in Figure 5B.

8) Ser210 may be the most frequently mutated residue in APT1 but at 5 mutations out of xxx tumors, its potential significance should be discussed in the context of mutation frequencies found in well-established drivers of melanoma such as BRAF or NRAS for example.

This is an excellent point and we agree that it is unlikely this mutation is plays a role in cancer like bona fide oncogenes such as BRAF and NRAS. Because it occurs at a higher frequency than the other APT1 mutations that all reside outside the dimer interface, we think that this mutation is beneficial to the primary tumor in vivothrough a mechanism that is not easily observed in cell culture. We added to the Discussion the idea that this mutation occurs at a low frequency compared to established drivers such as BRAF and NRAS and is therefore not likely an oncogenic driver.

“Not surprisingly, the S210L mutation in APT1 is observed at a much lower frequency compared to oncogenes frequently mutated in melanoma, such as BRAF or NRAS. We therefore think it is unlikely APT1 mutations contribute significantly to oncogenesis and tumor initiation.”

[Editors' note: further revisions were requested prior to acceptance, as described below.]

The manuscript has been improved but there are some remaining issues that need to be addressed before acceptance, as outlined below:The authors have strengthened the paper considerably with new experiments that support the model for the effect of Wnt5a on adhesion and invasiveness of tumor cells. These include comparison of the activity of Wnt3a with that of Wnt5a to bolster the argument that the effect is Wnt5a/non-canonical pathway specific; use of two general kinase inhibitors to further probe the key role of phosphorylation; demonstrating that inhibition of APT1 blocks the effect of the mutants; quantifying the effect of the S209D on enzyme activity; and presentation of data showing dimerization of the purified wt, but not mutant, protein.1) In the subsection “APT1 Phosphorylation Increases Depalmitoylating Activity”, it might be useful to explicitly note that the identity of the kinase is unknown.

We now explicitly note that the identity of the kinase is unknown:

“Because the identity of the kinase that phosphorylates APT1 is still unknown, we pre-treated WM239A cells expressing APT1^WT^ with either the serine/threonine kinase ATP-competitive inhibitor BI-D1870 or the broad-spectrum kinase inhibitor staurosporine for 1 hour, incubated the cells with the DPP-3 probe, and measured fluorescence over time by live-cell microscopy during treatment with Wnt5a or control buffer.”

2) The dimerization mechanism model requires further clarification and discussion:- The authors should be absolutely clear as to the basis of the statement "…phosphorylation of APT1 does not completely block dimerization…" Is it the very small rise at the highest concentration at 0.375 mg/ml SEC trace? The presence of "dimer" in the S209D gel in Figure 4D looks like an artifact of boiling in reducing sample buffer, which can cause some disulfide formation; in fact, C173 is surface exposed. Or is this inferred by the fact that the mutant retains some activity? If the latter, this is an interpretation based on the model and is something of a circular argument.

We included the statement that phosphorylation does not completely block dimerization to explain the crosslinking data and partly because of the small rise of the dimeric peak at the highest concentration. We agree the results are insufficient to say dimerization is not completely blocked by phosphorylation. Since we have chosen to remove the crosslinking data as per your comment below, we will also remove this statement to focus our conclusion on the inhibition of APT1 dimerization by phosphorylation.

- The model for dimerization assumes the dimer interface seen in the asymmetric unit of the crystal, which as noted in the original review is not compatible with the DSC crosslinking. The explanation provided by the authors is problematic, as it would imply transient formation of different dimers. There is another dimer in crystal lattice that could account for observed crosslinking, e.g. between K219 and K90. Perhaps this dimer could limit access of substrate, and electrostatic repulsion between D101 might explain S209D; or, there are allosteric effects of S209 and S210 on formation of this "alternative" dimer interface (this also touches on the analysis of the other mutations shown in Figure 5). In any case, either explicit discussion of the DSC result is needed or these crosslinking data should be removed. Without a definitive experimental analysis of the true dimer interface, the connection between activity and dimer formation remains correlative, and the effect of the mutants at S209 and S210 may well be due to other effects as they now note.

We agree the crosslinking data could have multiple interpretations. We would like to remove the crosslinking data and instead focus on the SEC data.

- The traces in Figure 4E should be labeled with the apparent MW of the standard peaks. Also, the fraction numbers are unreadable; perhaps colored bars indicating which fractions have been run on the gel would be better.

We have revised Figure 4E (Now Figure 4D) to include legible fraction numbers and we have labelled the figure with the apparent MW of the standard peaks.